# Neural Collapse in Multi-Task Learning

**Youjun Wang**[1]**, Boqi Li**[1]**, Xin Zou**[1]**, Weiwei Liu**[1] *
[1]Department of Computer Science, Wuhan University, Wuhan, China

## Abstract

Neural collapse (NC) plays a key role in understanding deep neural networks. However, existing empirical and theoretical studies of NC primarily focus on the single-task scenarios. This paper studies neural collapse in multi-task learning. We consider two standard feature-based multi-task learning scenarios: Single-Source Multi-Task Classification (SSMTC) and Multi-Source Multi-Task Classification (MSMTC). Interestingly, we find that the task-specific linear classifier and features converge to the Simplex Equiangular Tight Frame (ETF) in the setting of MSMTC. In the setting of SSMTC, task-specific linear classifier converges to the task-specific ETF and these task-specific ETFs are mutually orthogonal. Moreover, the shared features across tasks converge to the scaled sum of the weight vectors associated with the task-specific labels in each task's classifier. We also provide the theoretical guarantee for our empirical findings. Through detailed analysis, we uncover the mechanism of MTL where each task learns task-specific latent features that together form the shared features. Moreover, we reveal an inductive bias in MTL that task correlation reconfigures the geometry of task-specific classifiers and promotes alignment among the features learned by each task.

## 1 Introduction

Deep neural networks have achieved impressive performance in fields ranging from computer vision to natural language processing (Brown et al., 2020; Krizhevsky et al., 2012; Simonyan & Zisserman, 2015a; Vaswani et al., 2017). Recent work (Papyan et al., 2020) has revealed an intriguing phenomenon in deep neural networks, termed *Neural Collapse* (NC). When deep neural network training enters the terminal phase of training (TPT) (Papyan et al., 2020)—the stage where the training error reaches 0 and the training loss continues to decrease, the features and the last layer of classifiers exhibit a common pattern across different network architectures and datasets as shown in Figure 1(a) and Figure 1(d). This phenomenon can be summarized by the following four properties:

**(NC1) Variability collapse**. Each last-layer feature with the same class converges to its corresponding class mean.

**(NC2) Convergence to Simplex ETF**. The vectors of the class means (after centering by their global mean) converge to a Simplex ETF formally defined in Definition 3, achieving equal lengths, equal pairwise angles, and maximal distance in the feature space.

**(NC3) Convergence to self-duality**. Up to rescaling, the class-means and last-layer classifiers converge to each other.

**(NC4) Simplification to Nearest Class-Center**. The network classifier converges to selecting the class with the closest training class mean.

NC provides a mathematical characterization of features and last-layer classifier for deep neural network during TPT. NC has been observed and applied in various settings, such as imbalanced learning (Fang et al., 2021b; Thrampoulidis et al., 2022; Xie et al., 2023; Yan et al., 2024; Yang et al., 2022), transfer learning (Galanti et al., 2022; Li et al., 2022; Munn et al., 2024), continual learning (Yang et al., 2023; Yu et al., 2023), learning for a large number of classes (Jiang et al., 2024), multi-label learning (Li et al., 2024), regression (Andriopoulos et al., 2024), etc. More discussions about NC are included in Appendix E.1. Following previous work, the penultimate layers of the network

---

*Corresponding author, Email: liuweiwei863@gmail.com

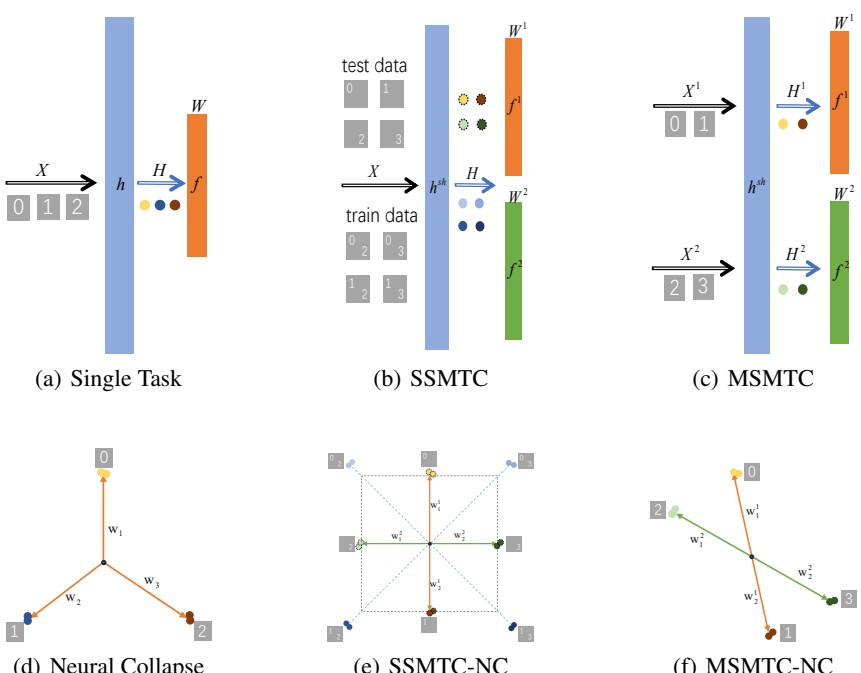

Figure 1: Illustration of single-task classification, single-source multi-task classification (SSMTC), and multi-source multi-task classification (MSMTC) settings (top row) and the corresponding neural collapse phenomenon (bottom row). The colored points represent features of samples from different classes. The arrows indicate the classifier weight vectors for each class. Figure 1(a) shows the single-task setting: a feature extractor $h$ and a classifier $f$. Figures 1(b) and 1(c) show the network structure in MTL, which consists of a shared feature extractor $h^{sh}$ and multiple task-specific classifiers $\{f^t\}$ ($t \in \{1, 2\}$). Figure 1(d) illustrates the geometric structure of the features and classifiers in the case of three-class classification. In MTL, we consider two binary classification tasks for clarity. As shown in Figure 1(e), the features of training samples with label $(0, 2)$ collapse to the scaled sum of $\boldsymbol{w}_1^1$ and $\boldsymbol{w}_1^2$. Interestingly, the features of test samples with single label 0 align with $\boldsymbol{w}_1^1$ and the features of test samples with single label 2 align with $\boldsymbol{w}_1^2$. Figure 1(f) shows that the task-specific features collapse to the scaled task-specific classifier. In Figure 1(e) and Figure 1(f), $\boldsymbol{w}_*^1$ and $\boldsymbol{w}_*^2$ both form the Simplex ETF where $* \in \{1, 2\}$. Moreover, $\boldsymbol{w}_*^1$ is orthogonal to $\boldsymbol{w}_*^2$ in SSMTC-NC as shown in Figure 1(e).

are considered as the feature extractor and the last layer of the network is referred to as the classifier. However, most empirical and theoretical studies on NC focus on the relationship between a single linear classifier and features in a single task, with limited attention given to multiple classifiers in multi-task learning (MTL). Our work investigates NC in the context of MTL, exploring the distinct latent features each task learns and how task correlation influences the latent features.

**Related Work on MTL**. Feature-based MTL approach is one of the most popular MTL methods, which aims to learn the common representations for multiple tasks (Zhang & Yang, 2022). Recent studies on feature-based MTL have predominantly focused on the biases introduced in the shared features learned across tasks (Collins et al., 2024; Lippl & Lindsey, 2024; Maurer et al., 2016; Shenouda et al., 2024; Wu et al., 2020). Our work focuses on the TPT, providing a geometric characterization of the feature space to offer a more nuanced understanding of MTL. More discussions about MTL are included in Appendix E.2. Following Rosenbaum et al. (2018); Sener & Koltun (2018); Shenouda et al. (2024); Yu et al. (2020), we consider two common scenarios in MTL according to different datasets: Single-Source Multi-Task Classification (SSMTC) and Multi-Source Multi-Task Classification (MSMTC). SSMTC focuses on classifying a single label for samples within each task. For example, each task classifies a label of an image in Multi-MNIST (LeCun et al., 2010). In contrast, MSMTC involves classifying data from distinct subsets of categories for

each task, such as in CIFAR100-Split-5x20 (Rosenbaum et al., 2018), where the dataset is divided into 20 tasks, with each task classifying 5 categories.

**Contributions**. As shown in Figure 1, we study the geometric properties of the learned features and the task-specific classifiers in MTL to provide a nuanced understanding of MTL. Our contributions can be summarized as follows:

- **Multi-Task Neural Collapse Phenomenon**. As shown in Figure 1, we study two standard settings of MTL: SSMTC (Figure 1(b)) and MSMTC (Figure 1(c)). The two settings exhibit different Simplex ETF structures, termed SSMTC-NC (Figure 1(e)) and MSMTC-NC (Figure 1(f)). We find that each task-specific classifier converges to the task-specific Simplex ETF in both SSMTC and MSMTC settings. Interestingly, we also find that the shared features across tasks converge to the scaled sum of weight vectors of the task-specific linear classifier for each task-specific label in SSMTC-NC. The task-specific last-layer features in MSMTC converge to their respective task-specific classifiers.

- **Global Optimality of SSMTC-NC and MSMTC-NC**. Theoretically, we prove that under the assumption of Unconstrained Feature Model (UFM) (Fang et al., 2021a; Mixon et al., 2022), any global optimal solution satisfies the properties of SSMTC-NC and MSMTC-NC.

- **Insights on MTL**. Through further detailed analysis, we find that shared features are formed by the features learned by each task in SSMTC. Moreover, we reveal a fundamental inductive bias in MTL that task correlation reshapes the space of task-specific classifiers, thereby promoting alignment among the features learned by each task.

## 2 PROBLEM FORMULATION

### 2.1 MULTI-TASK NEURAL NETWORK

Our multi-task architecture employs **hard parameter sharing** (Rosenbaum et al., 2018; Ruder, 2017) as the multi-task neural network structure composed of shared feature extraction layers $h^{sh}$ and task-specific linear classifiers $\{f^t\}$ ($t \in [T]$) where $T$ is the number of tasks. Given a sample $\boldsymbol{x}$, the output $g^t$ for the $t$-th task is:

$$g^t(\boldsymbol{x}) = f^t(h^{sh}(\boldsymbol{x}))$$

where

$$f^t(\boldsymbol{h}) = \boldsymbol{W}^t \boldsymbol{h} + \boldsymbol{b}^t$$

$$h^{sh}(\boldsymbol{x}) = \sigma(\boldsymbol{W}_{L-1} \ldots \sigma(\boldsymbol{W}_1 \boldsymbol{x} + \boldsymbol{b}_1) \ldots + \boldsymbol{b}_{L-1})$$

$\boldsymbol{W}^t$ represents the task-specific linear classifier of the last-layer and $h^{sh}(\boldsymbol{x})$ is the shared feature of the input $\boldsymbol{x}$ for all tasks. For a $L$-layer neural network, each layer consists of an affine transformation followed by a nonlinear activation function $\sigma(\cdot)$ (e.g.,ReLU). We use $\boldsymbol{\theta}^{sh}$ to denote shared parameters across all the tasks and $\boldsymbol{\theta}^t$ to denote task-specific parameters.

### 2.2 SINGLE-SOURCE MULTI-TASK CLASSIFICATION

We consider a MTL problem over an input space $\mathcal{X}$ and a collection of task spaces $\{\mathcal{Y}^t\}_{t \in [T]}$. Each task space $\mathcal{Y}^t$ contains $K_t$ distinct classes. The MTL dataset is $\{\boldsymbol{x}_i, y_i^1, \ldots, y_i^T\}_{i \in [N]}$ that are independent and identically distributed. $N$ is the number of samples, and $y_i^t$ is the label of the $t$-th task for the $i$-th data point. We denote $\boldsymbol{y_i} = (y_i^1, \ldots, y_i^T)$ the **label** for the sample $\boldsymbol{x}_i$ and $y_i^t$ the **task-specific label** in SSMTC. Consider a parametric hypothesis class per task as $g^t(\boldsymbol{x}; \boldsymbol{\theta}^{sh}, \boldsymbol{\theta}^t) : \mathcal{X} \to \mathcal{Y}^t$ and task-specific loss functions $\mathcal{L}^t(\cdot, \cdot) : \mathcal{Y}^t \times \mathcal{Y}^t \to \mathbb{R}^+$, we minimize the following empirical risk:

$$\min_{\substack{\boldsymbol{\theta}^{sh} \\ \boldsymbol{\theta}^1, \ldots, \boldsymbol{\theta}^T}} \sum_{t=1}^T c^t \hat{\mathcal{L}}^t \left( \boldsymbol{\theta}^{sh}, \boldsymbol{\theta}^t \right) \tag{1}$$

where $c^t$ denotes task-specific weights and $\hat{\mathcal{L}}^t(\boldsymbol{\theta}^{sh}, \boldsymbol{\theta}^t)$ denotes empirical loss of the task $t$, defined as:

$$\hat{\mathcal{L}}^t\left(\boldsymbol{\theta}^{sh}, \boldsymbol{\theta}^t\right) \triangleq \frac{1}{N} \sum_{i=1}^{N} \mathcal{L}\left(g^t\left(\boldsymbol{x}_i; \boldsymbol{\theta}^{sh}, \boldsymbol{\theta}^t\right), y_i^t\right)$$

## 2.3 MULTI-SOURCE MULTI-TASK CLASSIFICATION

The key difference between Single-Source Multi-Task Classification and Multi-Source Multi-Task Classification is that the latter has multiple input spaces $\{\mathcal{X}^t\}_{t \in [T]}$ and corresponding task space $\{\mathcal{Y}^t\}_{t \in [T]}$ where $T$ is the number of the tasks. Assume that each task has $N^t$ data points $\{\boldsymbol{x}_i^t, y_i^t\}_{i \in [N^t]}$ that are independent and identically distributed. Similarly, we denote the parametric hypothesis class $g^t(\boldsymbol{x}; \boldsymbol{\theta}^{sh}, \boldsymbol{\theta}^t) : \mathcal{X}^t \to \mathcal{Y}^t$ and the same task-specific loss functions $\mathcal{L}^t(\cdot, \cdot)$. Taking the task weights into account, we minimize the following empirical risk:

$$\min_{\substack{\boldsymbol{\theta}^{sh} \\ \boldsymbol{\theta}^1, \dots, \boldsymbol{\theta}^T}} \sum_{t=1}^{T} \frac{c^t}{N^t} \sum_{i=1}^{N^t} \mathcal{L}\left(g^t\left(\boldsymbol{x}_i^t; \boldsymbol{\theta}^{sh}, \boldsymbol{\theta}^t\right), y_i^t\right) \tag{2}$$

## 2.4 TRAINING OBJECTIVE FUNCTION UNDER UFM

**Unconstrained Feature Model (UFM)**. The UFM (Mixon et al., 2022) and layer-peeled model (Fang et al., 2021a), in which the features of the last layer are treated as free optimization variables, are commonly employed for the theoretical analysis of NC phenomena (Jiang et al., 2024; Súkeník et al., 2024; Xie et al., 2023; Yang et al., 2022). The rationale behind this model is that modern deep networks are extremely over-parameterized and expressive such that their feature mapping can be adapted to any training data (Zhang et al., 2021).

Following Jiang et al. (2024); Yang et al. (2022), we treat the feature $h^{sh}(x)$ as the free optimization variables. The training objective functions Eq.(1) and Eq.(2) can be transformed into the following Eq.(3) and Eq.(4) based on the UFM:

**Definition 1** (SSMTC Nonconvex Training Loss under UFM). *Let $\boldsymbol{W}^t \in \mathbb{R}^{K_t \times d}$, $\boldsymbol{b}^t \in \mathbb{R}^{K_t}$ be the task-specific weights and biases of last layer linear classifier, $\boldsymbol{H} = [\boldsymbol{h}_1; \boldsymbol{h}_2; \dots, \boldsymbol{h}_N] \in \mathbb{R}^{d \times N}$ be the feature matrix and $\boldsymbol{Y}^t$ be the one-hot encoding matrix for the $t$-th task. $d$ is the dimension of the feature. We consider the following optimization problem:*

$$\min_{\substack{\boldsymbol{W}^1, \boldsymbol{W}^2, \dots, \boldsymbol{W}^T \\ \boldsymbol{H}, \boldsymbol{b}^1, \dots, \boldsymbol{b}^T}} \sum_{t=1}^{T} c^t \mathcal{L}_{CE}(\boldsymbol{W}^t \boldsymbol{H} + \boldsymbol{b}^t, \boldsymbol{Y}^t) + \lambda_{\boldsymbol{H}} \|\boldsymbol{H}\|_F^2 + \lambda_{\boldsymbol{W}} \sum_{t=1}^{T} c^t \|\boldsymbol{W}^t\|_F^2 + \lambda_{\boldsymbol{b}} \sum_{t=1}^{T} c^t \|\boldsymbol{b}^t\|_2^2$$

(3)

*where $\lambda_{\boldsymbol{H}}$, $\lambda_{\boldsymbol{W}}$ and $\lambda_{\boldsymbol{b}}$ are the positive regularization coefficients and control the strength of the weight decay which prevents the norm of $\boldsymbol{W}^t$, $\boldsymbol{b}^t$ and $\boldsymbol{H}$ from growing to infinity. In this paper, $\|\cdot\|_F$ denotes the Frobenius norm and $\|\cdot\|_2$ denotes the Euclidean norm of the vector.*

**Definition 2** (MSMTC Nonconvex Training Loss under UFM). *Let $\boldsymbol{W}^t \in \mathbb{R}^{K_t \times d}$, $\boldsymbol{b}^t \in \mathbb{R}^{K_t}$ be the task-specific weights and biases of last layer linear classifier, $\boldsymbol{H}^t \in \mathbb{R}^{d \times N_t}$ be the task-specific feature matrix and $\boldsymbol{Y}^t$ be the one-hot encoding matrix for the $t$-th task. We consider the following optimization problem:*

$$\min_{\substack{\boldsymbol{W}^1, \boldsymbol{W}^2, \dots, \boldsymbol{W}^T \\ \boldsymbol{H}^1, \dots, \boldsymbol{H}^T, \boldsymbol{b}^1, \dots, \boldsymbol{b}^T}} \sum_{t=1}^{T} c^t \mathcal{L}_{CE}(\boldsymbol{W}^t \boldsymbol{H}^t + \boldsymbol{b}^t, \boldsymbol{Y}^t) + \sum_{t=1}^{T} c^t (\lambda_{\boldsymbol{H}} \|\boldsymbol{H}^t\|_F^2 + \lambda_{\boldsymbol{W}} \|\boldsymbol{W}^t\|_F^2 + \lambda_{\boldsymbol{b}} \|\boldsymbol{b}^t\|_2^2)$$

(4)

**Cross-entropy Loss**. In this work, we adopt the cross-entropy (CE) loss $\mathcal{L}_{CE}$.

**Task Weights**. Following Sener & Koltun (2018); Yu et al. (2020); Zhang & Yang (2022), we use the uniform task weight in this paper. We show the experimental results in Appendix D.3 of SSMTC-NC and MSMTC-NC when using MGDA (Sener & Koltun, 2018), Uncertainty Weight (Kendall et al., 2018), PCgrad (Yu et al., 2020), DWA (Liu et al., 2019a), FAMO (Liu et al., 2023), FairGrad (Ban & Ji, 2024) to update the task weights.

## 3 MAIN RESULTS

In this section, we present our empirical findings of SSMTC-NC and MSMTC-NC, and provide a theoretical guarantee.

### 3.1 SSMTC-NC

Following Fang et al. (2021a); Mixon et al. (2022); Zhu et al. (2021), we assume that the training data is balanced for each label.

$$\sum_{i=1}^{N}\prod_{t=1}^{T}\mathbb{I}[\boldsymbol{y}_i^t = k_t] = n, \forall k_1 \in [K_1], k_2 \in [K_2], \ldots, k_T \in [K_T]$$

In this section, for simplicity, we assume that the number of classes for each classification task is the same and use the uniform task weights (see Appendix B.2 for generalizations):

$$K_1 = K_2 = \ldots = K_T = K \tag{5}$$

$$c^1 = c^2 = \ldots = c^T = 1 \tag{6}$$

We denote $\boldsymbol{h}_j^{k_1,k_2,\ldots,k_T}$ the $j$-th feature whose label is $(k_1, k_2, \ldots, k_T)$, $\boldsymbol{\mu}^{k_1,k_2,\ldots,k_T}$ the mean of features with the corresponding label and $\boldsymbol{\mu}_G$ the mean of all features. $\boldsymbol{w}_k^t$ represents the classification weight for class $k$ in task $t$ and $\tilde{\boldsymbol{w}}_k^t$ denotes the $\boldsymbol{w}_k^t/\|\boldsymbol{w}_k^t\|_2$. $\tilde{\boldsymbol{h}}_j^{k_1,k_2,\ldots,k_T}$ denotes the $(\boldsymbol{h}_j^{k_1,k_2,\ldots,k_T} - \boldsymbol{\mu}_G)/\|\boldsymbol{h}_j^{k_1,k_2,\ldots,k_T} - \boldsymbol{\mu}_G\|_2$. The Kronecker delta, denoted as $\delta_{x,y}$, is a function used to indicate whether two variables $x$ and $y$ are equal. **Ave** represents the operation of taking the average. $\langle \cdot, \cdot \rangle$ denotes the dot product operation.

We use the objective function (3) to train the multi-task network. When we approach the TPT, we find the following experiment phenomena, which we refer to as SSMTC-NC:

**(NC1) Variability Collapse**. Each feature with the same label converges to the mean.

$$\underset{k_1\ldots,k_T,j}{\mathbf{Ave}}\{(\boldsymbol{h}_j^{k_1\ldots,k_T} - \boldsymbol{\mu}^{k_1\ldots,k_T})(\boldsymbol{h}_j^{k_1\ldots,k_T} - \boldsymbol{\mu}^{k_1\ldots,k_T})^{\mathsf{T}}\} \to 0$$

**(NC2) Task-specific Simplex ETF**. The weights of the classifier for each task form a Simplex ETF.

$$|\|\boldsymbol{w}_k^t\|_2 - \|\boldsymbol{w}_{k'}^t\|_2| \to 0, \forall k, k' \in [K], t \in [T]$$

$$\langle \tilde{\boldsymbol{w}}_k^t, \tilde{\boldsymbol{w}}_{k'}^t \rangle \to \frac{K}{K-1}\delta_{k,k'} - \frac{1}{K-1}, \forall k, k' \in [K], t \in [T]$$

**(NC3) Orthogonality**. The spaces spanned by the weights of the classifiers for any two tasks are mutually orthogonal.

$$\langle \tilde{\boldsymbol{w}}_k^t, \tilde{\boldsymbol{w}}_{k'}^{t'} \rangle \to 0, \forall k, k' \in [K], t, t' \in [T] \text{ with } t \neq t'$$

**(NC4) Convergence to Sum of Task-specific Classifiers**. The shared features among different tasks converge to the scaled sum of weight vectors of the task-specific linear classifier for each task-specific label.

$$\left\| \tilde{\boldsymbol{h}}_j^{k_1,k_2,\ldots,k_T} - \frac{\sum_{t=1}^{T}\boldsymbol{w}_{k_t}^t}{\|\sum_{t=1}^{T}\boldsymbol{w}_{k_t}^t\|_2} \right\|_2 \to 0$$

**(NC5) Simplification to Nearest Class-Center**. The network's classification result converges to choose whichever class has the nearest training class-mean.

$$(\underset{k_1}{\mathbf{argmax}}\langle\boldsymbol{w}_{k_1}^1, \boldsymbol{h}\rangle + b_{k_1}^1, \ldots, \underset{k_T}{\mathbf{argmax}}\langle\boldsymbol{w}_{k_T}^T, \boldsymbol{h}\rangle + b_{k_T}^T) \to \underset{k_1\ldots,k_T}{\mathbf{argmin}}\|\boldsymbol{h} - \boldsymbol{u}^{k_1\ldots,k_T}\|_2 \tag{7}$$

Theoretically, we rigorously analyze the global minimizers of Problem (3) and get the form of minimizers to validate the experimental results.

**Theorem 3.1.** *Assume that the training data is balanced, the feature dimension is larger than the number of classes, i.e., $d \geq \sum_{t=1}^{T} K_t - T$, the regularization parameters satisfy $\lambda_{\boldsymbol{H}} \lambda_{\boldsymbol{W}} < \frac{N}{4K}$, and the Assumptions (5), (6) hold, then, any global minimizers of Problem (3) satisfy the following five properties, which correspond to the five phenomena of SSMTC-NC (SSMTC-NC1 to SSMTC-NC5):*

1. $\boldsymbol{h}_j^{k_1...,k_T} = \boldsymbol{\mu}^{k_1...,k_T}$

2. $\|\boldsymbol{w}_k^t\|_2 = \|\boldsymbol{w}_{k'}^t\|_2, \langle \tilde{\boldsymbol{w}}_k^t, \tilde{\boldsymbol{w}}_{k'}^t \rangle = \frac{K}{K-1}\delta_{k,k'} - \frac{1}{K-1}, \forall k,k' \in [K], t \in [T]$

3. $\langle \tilde{\boldsymbol{w}}_k^t, \tilde{\boldsymbol{w}}_{k'}^{t'} \rangle = 0, \forall k,k' \in [K], t,t' \in [T]$ with $t \neq t'$

4. $\tilde{\boldsymbol{h}}_j^{k_1,k_2,...,k_T} = \frac{\sum_{t=1}^{T} \boldsymbol{w}_{k_t}^t}{\|\sum_{t=1}^{T} \boldsymbol{w}_{k_t}^t\|_2}$

5. $(\underset{k_1}{\operatorname{argmax}}\langle \boldsymbol{w}_{k_1}^1, \boldsymbol{h} \rangle + b_{k_1}^1, \ldots, \underset{k_T}{\operatorname{argmax}}\langle \boldsymbol{w}_{k_T}^T, \boldsymbol{h} \rangle + b_{k_T}^T) = \underset{k_1,...,k_T}{\operatorname{argmin}}\|\boldsymbol{h} - \boldsymbol{u}^{k_1...,k_T}\|_2$

The detailed proof of Theorem 3.1 is deferred to Appendix B. Next, we delve into the implications of our findings from various perspectives.

**Mutually Orthogonal Task-specific Classifiers**. The orthogonality in **NC3** ensures that the task-specific classifiers can be optimized independently.

**Feature Learning in MTL**. **NC3** and **NC4** suggest that the learned features of Problem (3) may be a combination of features from multiple orthogonal subspaces, which is further validated in the experimental section 5.1.

**General SSMTC-NC**. Considering a general training scenario without Assumptions (5) and (6), a similar SSMTC-NC phenomenon still exists. The detailed descriptions and theories are provided in Appendix B.2, and the experimental results are provided in Appendix D.2.

## 3.2 MSMTC-NC

We assume that the number of training samples in each class is balanced for each task and the feature dimension is greater than the largest number of classes in all tasks, i.e., $d \geq \max_t K_t - 1$. We use the objective function (4) to train the multi-task network and experimentally observe that the neural collapse phenomenon occurs for each task, referred to as **MSMTC-NC**.

**(NC1) Within-class Variability Collapse**. In each task, the features of the same class are clustered around the class mean.

$$\underset{k,j}{\operatorname{Ave}}(\boldsymbol{h}_{k,j}^t - \boldsymbol{\mu}_k^t)(\boldsymbol{h}_{k,j}^t - \boldsymbol{\mu}_k^t)^{\mathsf{T}} \to 0, \forall t \in [T]$$

**(NC2) Convergence to Simplex ETF**. The classifier and features of each task converge to a Simplex ETF.

$$\langle \tilde{\boldsymbol{w}}_k^t, \tilde{\boldsymbol{w}}_{k'}^t \rangle \to \frac{K_t}{K_t - 1}\delta_{k,k'} - \frac{1}{K_t - 1}, \forall t \in [T], k,k' \in [K_t]$$

$$\langle \tilde{\boldsymbol{\mu}}_k^t, \tilde{\boldsymbol{\mu}}_{k'}^t \rangle \to \frac{K_t}{K_t - 1}\delta_{k,k'} - \frac{1}{K_t - 1}, \forall t \in [T], k,k' \in [K_t]$$

$$|\|\boldsymbol{w}_k^t\|_2^2 - \|\boldsymbol{w}_{k'}^t\|_2^2| \to 0, \forall t \in [T], k,k' \in [K_t]$$

$$|\|\boldsymbol{\mu}_k^t\|_2^2 - \|\boldsymbol{\mu}_{k'}^t\|_2^2| \to 0, \forall t \in [T], k,k' \in [K_t]$$

**(NC3) Convergence to Self-duality**. The linear classifiers and class-means of each task will converge to align with each other.

$$\|\tilde{\boldsymbol{\mu}}_k^t - \tilde{\boldsymbol{w}}_k^t\|_2^2 \to 0, \forall t \in [T], k \in [K_t]$$

**(NC4) Simplification to Nearest Class-Center**.

$$\underset{k}{\operatorname{argmax}}\langle \boldsymbol{w}_k^t, \boldsymbol{h} \rangle + b_k^t \to \underset{k}{\operatorname{argmin}}\|\boldsymbol{h} - \boldsymbol{u}_k^t\|_2, \forall t \in [T]$$

$\boldsymbol{h}_{k,j}^t$ denotes the feature of the $j$-th sample belonging to class $k$ in the $t$-th task and $\boldsymbol{\mu}_k^t$ represents the mean, i.e., $\boldsymbol{\mu}_k^t = \sum_{j=1}^{n_t} \boldsymbol{h}_{k,j}^t / n_t$ ($n_t$ denotes the number of samples per class in the $t$-th task). $\tilde{\boldsymbol{\mu}}_k^t$ denotes $(\boldsymbol{\mu}_k^t - \boldsymbol{\mu}_G) / \|\boldsymbol{\mu}_k^t - \boldsymbol{\mu}_G\|_2$.

Theoretically, we rigorously analyze the global minimizers of Problem (4) and get the form of minimizers to validate the experimental results.

**Theorem 3.2.** *Assume that the feature dimension is larger than the largest number of classes in all tasks, i.e., $d \geq \max_t K_t - 1$, the regularization parameters satisfy $\lambda_H \lambda_W < \frac{n_t}{4}$, and the number of training samples in each class is balanced for each task, any global minimizers of Problem (4) satisfy the following four properties, which correspond to the four phenomena of MSMTC-NC (MSMTC-NC1 to MSMTC-NC4):*

1. $\boldsymbol{h}_{k,j}^t = \boldsymbol{\mu}_k^t, \forall t \in [T], k \in [K_t]$

2. $\|\boldsymbol{w}_k^t\|_2 = \|\boldsymbol{w}_{k'}^t\|_2, \langle \tilde{\boldsymbol{w}}_k^t, \tilde{\boldsymbol{w}}_{k'}^t \rangle = \frac{K}{K-1}\delta_{k,k'} - \frac{1}{K-1}, \forall t \in [T], k, k' \in [K_t]$

3. $\tilde{\boldsymbol{\mu}}_k^t = \tilde{\boldsymbol{w}}_k^t, \forall t \in [T], k \in [K_t]$

4. $\underset{k}{\operatorname{argmax}}\langle \boldsymbol{w}_k^t, \boldsymbol{h} \rangle + b_k^t = \underset{k}{\operatorname{argmin}}\|\boldsymbol{h} - \boldsymbol{u}_k^t\|_2, \forall t \in [T]$

A detailed proof of Theorem 3.2 is provided in Appendix B.3.

## 4 EXPERIMENT

### 4.1 DATASETS

The datasets used in this paper are Multi-MNIST (Sabour et al., 2017), Multi-CIFAR10, CIFAR100-Cross and CIFAR100-Split (Rosenbaum et al., 2018). The detailed description of the datasets is provided in Appendix C. Table 2 presents the datasets. It is worth mentioning that for SSMTC, CIFAR100-Cross-10x10 classification task can be viewed as multi-task learning with two tasks, each containing 10 classes. In contrast, for MSMTC, CIFAR100-Split-5x20 classification task involves multi-task learning with 20 tasks, each having 5 classes. More experiments on CelebA (Liu et al., 2015), ImageNet-1K (Deng et al., 2009) are shown in Appendix D.4.

### 4.2 EXPERIMENTAL SETTINGS

This paper uses ResNet18, ResNet34 (He et al., 2016), VGG11 and VGG13 (Simonyan & Zisserman, 2015b) as the shared feature extractor, and the uniform static weights. Throughout all the experiments, we use a SGD optimizer with fixed batch size 128, weight decay $(\lambda_H, \lambda_W) = (5 \cdot 10^{-4}, 5 \cdot 10^{-4})$ and momentum 0.9. In the experiment, the learning rate is initially set to $1 \cdot 10^{-1}$ and gradually decays to $1 \cdot 10^{-3}$ over 500 epochs using the CosineAnnealingLR scheduler.

Additionally, we employ different multi-task weighting strategies as shown in Appendix D.3, including MGDA, Uncertainty Weight, PCgrad, DWA, FAMO, FairGrad. And we also consider the datasets with different number of classes for each task, as shown in Appendix D.2. Experiments on different learning rate and $(\lambda_H, \lambda_W)$ are shown in Appendix D.5.

### 4.3 EVALUATION METRICS

**SSMTC-NC**. Following Papyan et al. (2020); Súkeník et al. (2024), we evaluate NC1 using within-class variance, denoted as SNC1. NC2 assesses the proximity of the classifier to the Simplex ETF using the angle and norm, denoted as SNC2-1 and SNC2-2. NC3 is assessed by the maximum absolute value of the cosine between classifiers of different tasks, denoted as SNC3. NC4 is evaluated using the difference between the normalized feature mean and the scaled sum of task-specific classifiers, denoted as SNC4. NC5 is evaluated using the error rate of classification based on Nearest Class-Center, denoted as SNC5.

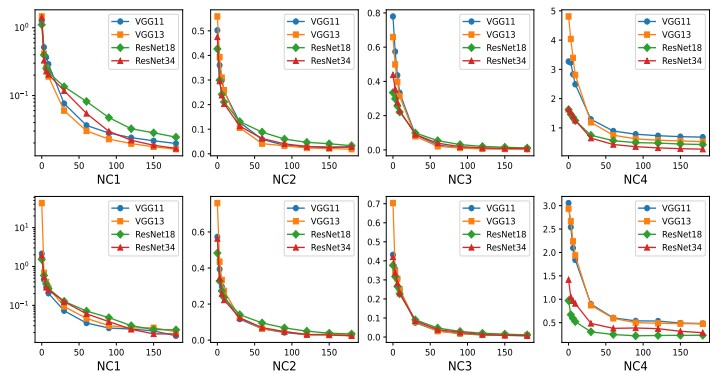 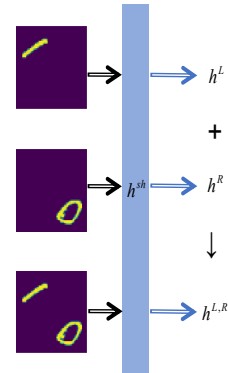

(a) NC on Multi-MNIST-10-10 (top) and Multi-CIFAR10-10-10 (bottom). The NC metrics for NC1, NC2, and NC3 are the means of the NC metrics on two tasks. The horizontal axis represents the epoch.

(b) Task-specific latent feature and shared feature

Figure 2: Illustration of Feature Learning in SSMTC

**MSMTC-NC**. Following Papyan et al. (2020); Súkeník et al. (2024), we use the mean of all tasks' metrics similar to SSMTC-NC and name them MNC1, MNC2-1, MNC2-2, MNC3 and MNC4 respectively.

The specific metric form is included in the Appendix D.9.

### 4.4 VERIFICATION OF SSMTC-NC AND MSMTC-NC

As shown in Figures 6, 7 and Figures 8, 9 in the Appendix, all the metrics approach zero during the terminal phase of training. These results show that SSMTC-NC exists in different networks and datasets. Figures 10, 11 in the Appendix show that MSMTC-NC exists in different networks and datasets. Building on prior works (Li et al., 2024; Yang et al., 2022), we conduct parameter-efficient training experiments leveraging NC properties in MTL as shown in Appendix D.7. Our results demonstrate the presence of both SSMTC-NC and MSMTC-NC phenomena, and verify the replaceability of final-layer parameters.

## 5 INSIGHTS FROM NC IN MTL

### 5.1 THE RELATIONSHIP BETWEEN SHARED FEATURES AND TASK-SPECIFIC LATENT FEATURES

To further understand the shared features and what each task learns respectively in SSMTC, we process the training samples $\boldsymbol{x}^{L,R}$ from the Multi-MNIST-10-10 and Multi-CIFAR10-10-10 datasets, retaining only the top-left and bottom-right features to obtain $\boldsymbol{x}^L$ and $\boldsymbol{x}^R$, respectively. We then feed test samples $\boldsymbol{x}^L$, $\boldsymbol{x}^R$ into the neural network trained on $\boldsymbol{x}^{L,R}$, to obtain $\boldsymbol{h}^L$ and $\boldsymbol{h}^R$ as shown in Figure 2(b). As shown in Figure 2(a), we perform NC analysis on $\boldsymbol{h}^L$ and the classifier $\boldsymbol{W}^L$, as well as on $\boldsymbol{h}^R$ and the classifier $\boldsymbol{W}^R$. We observe that both $\boldsymbol{h}^L$ and $\boldsymbol{h}^R$ exhibit the NC properties, including within-class variability collapse (NC1), convergence to Simplex ETF (NC2), and convergence to self-duality (NC3). We also uncover properties corresponding to SSMTC-NC4:

**(NC4) Convergence to Sum of Task-Specific Latent Features**

$$\left\| \tilde{\boldsymbol{\mu}}_{k_1,k_2}^{L,R} - \frac{\tilde{\boldsymbol{\mu}}_{k_1}^{L} + \tilde{\boldsymbol{\mu}}_{k_2}^{R}}{\| \tilde{\boldsymbol{\mu}}_{k_1}^{L} + \tilde{\boldsymbol{\mu}}_{k_2}^{R} \|} \right\|_2 \to 0$$

$\tilde{\boldsymbol{\mu}}_{k_1,k_2}^{L,R}$ denotes the feature mean of sample $\boldsymbol{x}^{L,R}$ with label $(k_1, k_2)$ after centering and normalization. $\tilde{\boldsymbol{\mu}}_{k_1}^{L}$ and $\tilde{\boldsymbol{\mu}}_{k_2}^{R}$ denote the feature means of sample $\boldsymbol{x}^L$ with class $k_1$ and sample $\boldsymbol{x}^R$ with class $k_2$ after centering and normalization.

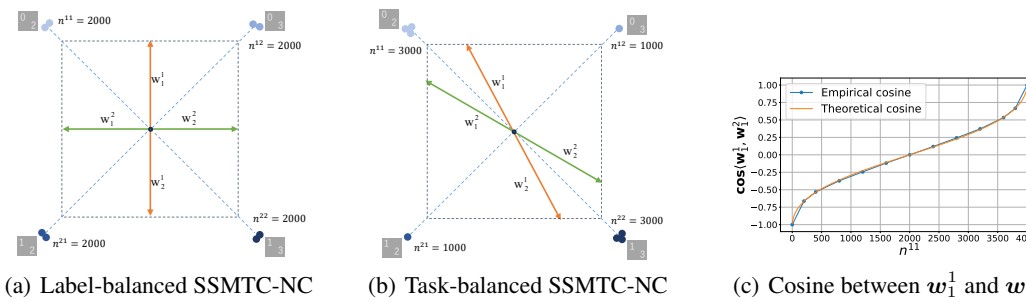

(a) Label-balanced SSMTC-NC     (b) Task-balanced SSMTC-NC     (c) Cosine between $\boldsymbol{w}_1^1$ and $\boldsymbol{w}_1^2$

Figure 3: Illustration of SSMTC-NC in the label-balanced case, SSMTC-NC in the task-balanced case, and cosine between $\boldsymbol{w}_1^1$ and $\boldsymbol{w}_1^2$. $n^{k_1 k_2}$ denote the number of samples with label $(k_1, k_2)$.

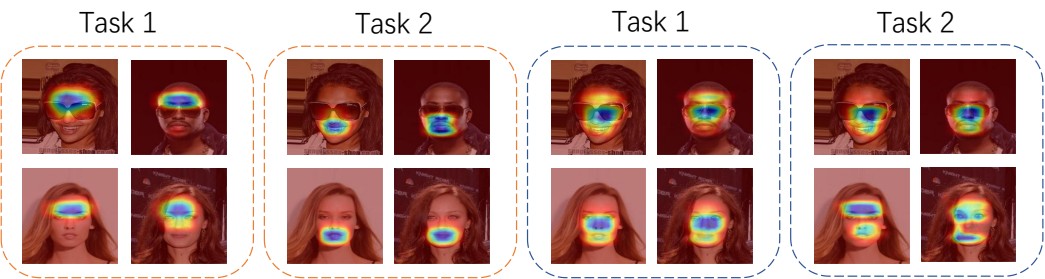

Figure 4: Grad-CAM++ visualizations for two tasks under independent and related settings. Each dashed block contains one example of each label. The first two blocks show Task 1 and Task 2 under independence ($\mathbf{cos}(\boldsymbol{w}_1^1, \boldsymbol{w}_1^2) = 0$), while the third and fourth blocks show Task 1 and Task 2 when tasks are related ($\mathbf{cos}(\boldsymbol{w}_1^1, \boldsymbol{w}_1^2) = 0.5$). Blue regions indicate the important regions for the task to predict the concept in the image.

NC1, NC2, and NC3 indicate that $\boldsymbol{h}^L$ and $\boldsymbol{h}^R$ are the features learned by the two tasks, which we refer to as **task-specific latent features**. NC4 further explains that the shared features of samples in MTL are composed of task-specific latent features. The above findings dissect the specific structure of the shared feature, indicating that task-specific classifiers decompose the shared features into task-specific latent features.

## 5.2 IMPACT OF TASK CORRELATION ON TASK-SPECIFIC CLASSIFIERS

With the total sample size fixed at $N$, we systematically vary the sampling proportions of each label pair $(k_1, k_2)$ and investigate how task correlation (Li et al., 2025; Ma et al., 2018) impacts the task-specific classifiers. Through empirical experiments and theoretical analysis, we find that, when moving from the label-balanced to the task-balanced scenario, all other NC properties continue to hold and SSMTC-NC3 adapts to Correlated-NC:

**(NC1) Variability collapse**. Each feature with the same label converges to the mean.

**(NC2) Task-specific Simplex ETF**. The weights of the classifier for each task form a Simplex ETF.

**(Correlated-NC3) Alignment of task-specific classifiers**. As $n^{k_1, k_2}$ increases, $\mathbf{cos}(\boldsymbol{w}_{k_1}^1, \boldsymbol{w}_{k_2}^2)$ increases.

**(NC4) Convergence to weighted sum of task-specific classifiers**.

**(NC5) Simplification to Nearest Class-Center**.

**Theorem 5.1.** *Consider two binary classification tasks, the training data is task-balanced, and the number of samples with label $(k_1, k_2)$ is $n^{k_1, k_2} (k_1 \in [2], k_2 \in [2])$. Assume that the number of all samples is $N$ and $\lambda_{\boldsymbol{H}} \lambda_{\boldsymbol{W}} < \frac{n^{k_1, k_2}}{2}$, then any global minimizers of Problem (3) satisfies the following conditions:*

1. $\boldsymbol{h}_j^{k_1,k_2} = \boldsymbol{\mu}^{k_1,k_2}$

2. $\|\boldsymbol{w}_k^t\|_2 = \|\boldsymbol{w}_{k'}^t\|_2, \langle \tilde{\boldsymbol{w}}_k^t, \tilde{\boldsymbol{w}}_{k'}^t \rangle = 2\delta_{k,k'} - 1, \forall k, k' \in [2], t \in [2]$

3. *As $n^{k_1,k_2}$ increases, $\cos(\boldsymbol{w}_{k_1}^1, \boldsymbol{w}_{k_2}^2)$ increases.*

4. $\tilde{\boldsymbol{h}}_j^{k_1,k_2} = \frac{\boldsymbol{w}_{k_1}^1 + \boldsymbol{w}_{k_2}^2}{\|\boldsymbol{w}_{k_1}^1 + \boldsymbol{w}_{k_2}^2\|_2}$

5. $\left(\underset{k_1}{\text{argmax}}\langle \boldsymbol{w}_{k_1}^1, \boldsymbol{h} \rangle + b_{k_1}^1, \underset{k_2}{\text{argmax}}\langle \boldsymbol{w}_{k_2}^2, \boldsymbol{h} \rangle + b_{k_2}^2\right) = \underset{k_1,k_2}{\text{argmin}}\|\boldsymbol{h} - \boldsymbol{u}^{k_1,k_2}\|_2$

The proof is shown in Appendix B.4. As shown in Figure 3(a) and Figure 3(b), we set $N = 8000$, $K = 2$, and then adjust the number of samples belonging to each label. As shown in Figure 3(c), when the number of samples with the label $(0, 2)$ increases, the cosine between $\boldsymbol{w}_1^1$ and $\boldsymbol{w}_1^2$ increases. We explain the phenomenon theoretically and derive the closed-form expression for $\cos(\boldsymbol{w}_1^1, \boldsymbol{w}_1^2)$ in Appendix B.4, which aligns with the experimental results as shown in Figure 3(c). The additional experimental results, encompassing extended scenarios with $K \geq 2$, are included in Appendix D.6.

Through the detailed experiments and theoretical analysis, we find that task correlation reshapes the space of task-specific classifiers.

**Implication of the alignment of task-specific classifiers**. When tasks are correlated, the alignment of task-specific classifiers implies that the features learned by the two tasks are aligned and closer to shared features. To verify the implication, we choose the `Eyeglasses` and `Mouth_Slightly_Open` attributes on CelebA as Task 1 and Task 2, and boost task correlation. We use Grad-CAM++ (Chattopadhyay et al., 2018) to visualize the important regions for the two tasks to predict the concept in the image. As shown in Figure 4, when tasks are related, their salient regions largely coincide, demonstrating that the two tasks attend to the same facial features and thus learn closely aligned representations.

## 6 CONCLUSION

In this work, we demonstrate neural collapse in MTL through SSMTC-NC and MSMTC-NC. We empirically validate both SSMTC-NC and MSMTC-NC across various network architectures and datasets. Theoretically, we prove that any global optimal solution must satisfy the properties of SSMTC-NC and MSMTC-NC. Through further experiments, we find that each task learns task-specific latent features that form shared features. Our empirical findings, supported by theoretical analysis, indicate that MTL is inherently biased toward leveraging task correlation to reconfigure the geometry of task-specific classifiers and promote alignment among the features learned by each task. We believe our work offers valuable insights into the understanding of features and classifiers in MTL.

## 7 ACKNOWLEDGMENTS

This work is supported by the Key R&D Program of Hubei Province under Grant 2024BAB038, the National Key R&D Program of China under Grant 2023YFC3604702.

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

# Appendix

APPENDIX ORGANIZATION

Table 1: Appendix Contents

| Appendix A | Basic Lemmas |
|---|---|
| Appendix B | Theoretical Proofs |
| Appendix C | Dataset Illustration and Visualization |
| Appendix D | Detailed Experimental Results |
| Appendix E | Discussions about the Related Work |

In Appendix A, we provide the lemmas used in the proofs. Appendices B.1 and Appendix B.3 present the proofs of Theorems 3.1 and Theorems 3.2, respectively, and Appendix B.2 offers the description and theoretical proof of the general SSMTC-NC. In Appendix B.4, we provide the proof for Theorem 5.1. Appendix C provides a detailed description of the datasets. Appendix D.1 presents other metrics. Appendix D.2 contains supplementary experiments for general SSMTC and MSMTC, and Appendix D.3 provides empirical results of SSMTC-NC and MSMTC-NC when we use multiple MTL methods to update the task weights. In Appendix D.4, we provide the experimental results on ImageNet, CelebA and Tiny-ImageNet. In Appendix D.5, we run the experiments on different learning rates and $(\lambda_H, \lambda_W)$. In Appendix D.6, we show the experimental results when the train data is task-balanced. In Appendix D.7, we provide the experimental results on efficient training utilizing NC properties in MTL. In Appendix E.1, we discuss the difference between Li et al. (2024) and our work. In Appendix E.2, we discuss the connection between our work and related work in MTL.

## A  BASIC DEFINITION AND LEMMAS

**Definition 3** (Simplex ETF (Papyan et al., 2020))**.** *A collection of vectors $m_i \in \mathbb{R}^d, i = 1, 2 \ldots, K, d \geq K - 1$, is said to be a Simplex equiangular tight frame if:*

$$\boldsymbol{M} = \sqrt{\frac{K}{K-1}} \boldsymbol{U}(\boldsymbol{I}_K - \frac{1}{K}\mathbf{1}_K\mathbf{1}_K^\mathsf{T}) \tag{8}$$

*where $\boldsymbol{M} = [\boldsymbol{m}_1, \ldots, \boldsymbol{m}_K] \in \mathbb{R}^{d \times K}$, $\boldsymbol{U} \in \mathbb{R}^{d \times K}$ allows a rotation and satisfies $\boldsymbol{U}^\mathsf{T}\boldsymbol{U} = \boldsymbol{I}_K$, $\boldsymbol{I}_K$ is the identity matrix, and $\mathbf{1}_K$ is the all-ones vector. Specially, when $K = 2$, the angle between any two vectors in $\boldsymbol{M}$ is $180$ degrees; When $K = 3$, the angle between any two vectors is $120$ degrees.*

**Lemma A.1.** *For a set of $n$ vectors $\boldsymbol{v}_i \in \mathbb{R}^d$, the point $\boldsymbol{x} \in \mathbb{R}^d$ that minimizes the sum of squared Euclidean distances*

$$\sum_{i=1}^n \|\boldsymbol{v}_i - \boldsymbol{x}\|_2^2$$

*is the mean of the vectors, i.e.,*

$$\boldsymbol{x} = \frac{1}{n}\sum_{i=1}^n \boldsymbol{v}_i.$$

**Lemma A.2.** *Let $f : \mathbb{R}^d \to \mathbb{R}$ be a strictly convex function. Then, for any $\boldsymbol{x}_1, \boldsymbol{x}_2, \ldots, \boldsymbol{x}_n \in \mathbb{R}^d$, the following inequality holds:*

$$\sum_{i=1}^n f(\boldsymbol{x}_i) \geq n \cdot f\left(\frac{1}{n}\sum_{i=1}^n \boldsymbol{x}_i\right).$$

*Equality holds if and only if $\boldsymbol{x}_1 = \boldsymbol{x}_2 = \cdots = \boldsymbol{x}_n$.*

**Lemma A.3.** *Consider the function $f(x) = \alpha \mathbf{log}(1 + \beta \mathbf{exp}(-\gamma x)) + x$, where $\alpha, \beta, \gamma \in \mathbb{R}^+$, and $x \in \mathbb{R}_{\geq 0}$. The minimizer $x^*$ of $f(x)$ is determined as follows:*

*- If $\beta + 1 \geq \alpha\beta\gamma$, then $x^* = 0$.*
*- Otherwise, $x^* = \frac{1}{\gamma} \mathbf{log}(\alpha\beta\gamma - \beta)$.*

# B    THEORETICAL PROOFS

**Remark.** In the SSMTC setting, all tasks share data drawn from the same source, and thus the coupling between tasks remains explicitly preserved through shared features. In contrast, in MSMTC, the optimization objective can indeed be decomposed into a sum of single-task objectives. This decomposition reflects that the effect of parameter sharing becomes negligible, but it does not mean that the model becomes multiple independent models; the analysis is still conducted within a single multi-task model. Therefore, within the UFM framework, the influence of parameter sharing is inherently stronger in SSMTC than in MSMTC. The effect of parameter sharing disappears in MSMTC. The key reasons are twofold: (1) the multi-task model has high representation capacity (UFM model), and (2) the data are drawn from multiple distinct sources (the MSMTC setting).Under these two conditions, the shared features can freely adjust to each task independently, which allows the optimization problem to decouple into task-specific subproblems. This decomposition holds in MSMTC but not in SSMTC. As a result, the theoretical results and experimental results observed in MSMTC closely resembles that in single task learning.

## B.1    PROOF OF THEOREM 3.1

The key idea of the proof of Theorem 3.1 is to lower bound the entropy loss in Problem (3). The equality holds only if SSMTC-NC is satisfied.

$$
\begin{aligned}
&g(\boldsymbol{W}^1, \boldsymbol{W}^2, \ldots, \boldsymbol{W}^T, \boldsymbol{H}, \boldsymbol{b}^1, \boldsymbol{b}^2, \ldots, \boldsymbol{b}^T) \\
&= \sum_{t=1}^{T} \sum_{k=1}^{K} \sum_{i=1}^{\frac{N}{K}} \mathbf{log}(1 + \sum_{l=1, l \neq k}^{K} \mathbf{exp}(\boldsymbol{w}_l^t \boldsymbol{h}_{k,i}^t - \boldsymbol{w}_k^t \boldsymbol{h}_{k,i}^t + b_l^t - b_k^t)) \\
&\quad + \lambda_{\boldsymbol{H}} \|\boldsymbol{H}\|_F^2 + \lambda_{\boldsymbol{W}} \sum_{t=1}^{T} \|\boldsymbol{W}^t\|_F^2 + \lambda_{\boldsymbol{b}} \sum_{t=1}^{T} \|\boldsymbol{b}^t\|_2^2
\end{aligned}
\tag{9}
$$

where $\boldsymbol{h}_{k,i}^t$ represents the $i$-th sample of label $k$ in the $t$-th task. First, we can find that the margin will not change if we subtract a vector $\boldsymbol{x}$ for all $\boldsymbol{w}_k^t$ or we subtract a scalar $y$ for all $b_l^t$ provided that $t$ is fixed. From Lemma A.1, subtracting the mean minimizes the regularization loss. So, the minimizer must satisfy:

$$
\begin{aligned}
\sum_{k=1}^{K} \boldsymbol{w}_k^t &= \boldsymbol{0}, \forall t \in [T] \\
\sum_{k=1}^{K} b_k^t &= 0, \forall t \in [T]
\end{aligned}
\tag{10}
$$

Observe that

$$\sum_{t=1}^{T}\sum_{k=1}^{K}\sum_{i=1}^{\frac{N}{K}}\log(1+\sum_{l=1,l\neq k}^{K}\exp(\boldsymbol{w}_l^t\boldsymbol{h}_{k,i}^t-\boldsymbol{w}_k^t\boldsymbol{h}_{k,i}^t+b_l^t-b_k^t))+\lambda_{\boldsymbol{H}}\|\boldsymbol{H}\|_F^2+\lambda_{\boldsymbol{W}}\sum_{t=1}^{T}\|\boldsymbol{W}^t\|_F^2+\lambda_{\boldsymbol{b}}\sum_{t=1}^{T}\|\boldsymbol{b}^t\|_2^2$$

$$\overset{(a)}{\geq}\sum_{t=1}^{T}\sum_{k=1}^{K}\sum_{i=1}^{\frac{N}{K}}\log(1+(K-1)\exp(-\frac{K}{K-1}(\boldsymbol{w}_k^t\boldsymbol{h}_{k,i}^t+b_k^t)))+\lambda_{\boldsymbol{H}}\|\boldsymbol{H}\|_F^2+\lambda_{\boldsymbol{W}}\sum_{t=1}^{T}\|\boldsymbol{W}^t\|_F^2+\lambda_{\boldsymbol{b}}\sum_{t=1}^{T}\|\boldsymbol{b}^t\|_2^2$$

$$\text{(a)}$$

$$\overset{(b)}{\geq}\sum_{t=1}^{T}\sum_{k=1}^{K}\frac{N}{K}\log(1+(K-1)\exp(-\frac{K^2}{(K-1)N}(\boldsymbol{w}_k^t\sum_{i=1}^{\frac{N}{K}}\boldsymbol{h}_{k,i}^t+\frac{N}{K}b_k^t)))$$

$$+\lambda_{\boldsymbol{H}}\|\boldsymbol{H}\|_F^2+\lambda_{\boldsymbol{W}}\sum_{t=1}^{T}\|\boldsymbol{W}^t\|_F^2+\lambda_{\boldsymbol{b}}\sum_{t=1}^{T}\|\boldsymbol{b}^t\|_2^2 \qquad \text{(b)}$$

$$\overset{(c)}{\geq}\sum_{t=1}^{T}N\log(1+(K-1)\exp(-\frac{K}{(K-1)N}\sum_{k=1}^{K}\boldsymbol{w}_k^t\sum_{i=1}^{\frac{N}{K}}\boldsymbol{h}_{k,i}^t))+\lambda_{\boldsymbol{H}}\|\boldsymbol{H}\|_F^2+\lambda_{\boldsymbol{W}}\sum_{t=1}^{T}\|\boldsymbol{W}^t\|_F^2+\lambda_{\boldsymbol{b}}\sum_{t=1}^{T}\|\boldsymbol{b}^t\|_2^2$$

$$\text{(c)}$$

$$\overset{(d)}{\geq}TN\log(1+(K-1)\exp(-\frac{K}{(K-1)NT}\sum_{k_1=1}^{K}\cdots\sum_{k_T=1}^{K}\sum_{i=1}^{n}\boldsymbol{h}_i^{k_1,...,k_T}\sum_{j=1}^{T}\boldsymbol{w}_{k_j}^j))+\lambda_{\boldsymbol{H}}\|\boldsymbol{H}\|_F^2+\lambda_{\boldsymbol{W}}\sum_{t=1}^{T}\|\boldsymbol{W}^t\|_F^2$$

$$\text{(d)}$$

$$\overset{(e)}{\geq}TN\log(1+(K-1)\exp(-\frac{K}{(K-1)NT}\sqrt{\frac{N}{4K\lambda_{\boldsymbol{H}}\lambda_{\boldsymbol{W}}}}(\lambda_{\boldsymbol{H}}\|\boldsymbol{H}\|_F^2+\lambda_{\boldsymbol{W}}\sum_{t=1}^{T}\|\boldsymbol{W}^t\|_F^2)))$$

$$+\lambda_{\boldsymbol{H}}\|\boldsymbol{H}\|_F^2+\lambda_{\boldsymbol{W}}\sum_{t=1}^{T}\|\boldsymbol{W}^t\|_F^2 \qquad \text{(e)}$$

Invoking Lemma A.3 with $\alpha=TN$, $\beta=K-1$, $\gamma=\frac{K}{(K-1)NT}\sqrt{\frac{N}{4K\lambda_{\boldsymbol{H}}\lambda_{\boldsymbol{W}}}}$ and $x=\lambda_{\boldsymbol{H}}\|\boldsymbol{H}\|_F^2+\lambda_{\boldsymbol{W}}\sum_{t=1}^{T}\|\boldsymbol{W}^t\|_F^2$, we derive that if $\lambda_{\boldsymbol{H}}\lambda_{\boldsymbol{W}}\geq\frac{N}{4K}$, the minimizer is $(\boldsymbol{W}^1,\ldots,\boldsymbol{W}^T,\boldsymbol{H})=(\boldsymbol{0},\ldots,\boldsymbol{0},\boldsymbol{0})$ and otherwise, the minimizer must obey $x^*>0$. Next, we examine the conditions required to reach the minimum value. In inequalities (a), (b), (c) and (d), we use Lemma A.2. The conditions under which equality holds are as follows:

$$\forall l_1,l_2\in[K]\setminus\{k\}, \boldsymbol{w}_{l_1}^t\boldsymbol{h}_{k,i}^t+b_{l_1}^t=\boldsymbol{w}_{l_2}^t\boldsymbol{h}_{k,i}^t+b_{l_2}^t$$

$$\forall i_1,i_2\in[\frac{N}{K}], \boldsymbol{w}_k^t\boldsymbol{h}_{k,i_1}^t=\boldsymbol{w}_k^t\boldsymbol{h}_{k,i_2}^t$$

$$\forall k_1,k_2\in[K], \boldsymbol{w}_{k_1}^t\sum_{i=1}^{\frac{N}{K}}\boldsymbol{h}_{k_1,i}^t+\frac{N}{K}b_{k_1}^t=\boldsymbol{w}_{k_2}^t\sum_{i=1}^{\frac{N}{K}}\boldsymbol{h}_{k_2,i}^t+\frac{N}{K}b_{k_2}^t \qquad (11)$$

$$\forall t_1,t_2\in[T], \sum_{k=1}^{K}\boldsymbol{w}_k^{t_1}\sum_{i=1}^{\frac{N}{K}}\boldsymbol{h}_{k,i}^{t_1}=\sum_{k=1}^{K}\boldsymbol{w}_k^{t_2}\sum_{i=1}^{\frac{N}{K}}\boldsymbol{h}_{k,i}^{t_2}$$

Inequality (d) is also based on $\lambda_{\boldsymbol{b}}\sum_{t=1}^{T}\|\boldsymbol{b}^t\|_2^2\geq 0$ and the data is balanced for each label. The number of samples for each label is $n$. Inequality (d) holds with equality iff

$$\forall t\in[T],k\in[K],b_k^t=0 \qquad (12)$$

Inequality (e) is based on

$$\boldsymbol{h}_i^{k_1,...,k_T}\sum_{j=1}^{T}\boldsymbol{w}_{k_j}^j\leq\lambda\|\boldsymbol{h}_i^{k_1,...,k_T}\|_2^2+\frac{1}{4\lambda}\|\sum_{j=1}^{T}\boldsymbol{w}_{k_j}^j\|_2^2 \qquad (13)$$

where $\lambda = \sqrt{\frac{N\lambda_H}{4K\lambda_W}}$ and the condition for this equality to be true is:

$$\forall i \in [n], \boldsymbol{h}_i^{k_1,\ldots,k_T} = \sqrt{\frac{K\lambda_W}{N\lambda_H}} \sum_{j=1}^{T} \boldsymbol{w}_{k_j}^{j} \tag{14}$$

Due to the condition (14), the optimizer must satisfy **NC1**, **NC4**. Because of the condition (12), we can remove bias in condition (11). By combining the condition (10), (11) and (14), we can conclude that:

$$\forall k_1, k_2 \in [K], t_1, t_2 \in [T], t_1 \neq t_2, \boldsymbol{w}_{k_1}^{t_1} \boldsymbol{w}_{k_2}^{t_2} = 0 \tag{15}$$

which satisfy **NC3**. Then we can infer that the optimizer must satisfy **NC2**, **NC5**. When one of the conditions is not met, the loss will be strictly greater than the minimum value. When one of the conditions is not met, the loss will be strictly greater than the minimum value, so in summary, combining all the conditions for equality, we can conclude that the optimal solution must satisfy the SSMTC-NC.

## B.2 GENERAL SSMTC-NC

When Assumption (5) and Assumption (6) are not satisfied, during the terminal phase of training using loss function in Problem (3), the features and classifiers of the network exhibit the following charactistics.

**(NC1) Variability collapse**. Each feature with the same label converges to the mean.

$$\underset{k_1\ldots,k_T,j}{\mathbf{Ave}}\{(\boldsymbol{h}_j^{k_1\ldots,k_T} - \boldsymbol{\mu}^{k_1\ldots,k_T})(\boldsymbol{h}_j^{k_1\ldots,k_T} - \boldsymbol{\mu}^{k_1\ldots,k_T})^{\mathsf{T}}\} \to 0$$

**(NC2) Task-specific Simplex ETF**. The weights of the classifier for each task form a Simplex ETF.

$$|\|\boldsymbol{w}_k^t\|_2 - \|\boldsymbol{w}_{k'}^t\|_2| \to 0, \forall k, k' \in [K_t], t \in [T]$$

$$\langle \tilde{\boldsymbol{w}}_k^t, \tilde{\boldsymbol{w}}_{k'}^t \rangle \to \frac{K_t}{K_t - 1}\delta_{k,k'} - \frac{1}{K_t - 1}, \forall k, k' \in [K_t], t \in [T]$$

**(NC3) Orthogonality**. The spaces spanned by the weights of the classifiers for any two tasks are mutually orthogonal.

$$\langle \tilde{\boldsymbol{w}}_k^t, \tilde{\boldsymbol{w}}_{k'}^{t'} \rangle \to 0, \forall k \in [K_t], k' \in [K_{t'}], t, t' \in [T] \text{ with } t \neq t'$$

**(NC4) Convergence to weighted sum of task-specific classifiers**. The shared features among different tasks converges to the weighted sum of task-specific linear classifier weight vectors for each task-specific label.

$$\left\| \tilde{\boldsymbol{h}}_j^{k_1,k_2,\ldots,k_T} - \frac{\sum_{t=1}^{T} \sqrt{c^t K_t}\boldsymbol{w}_{k_t}^t}{\|\sum_{t=1}^{T} \sqrt{c^t K_t}\boldsymbol{w}_{k_t}^t\|_2} \right\|_2 \to 0$$

**(NC5) Simplification to Nearest Class-Center**. The network's classification result converges to choose whichever class has the nearest training class-mean.

$$(\underset{k_1}{\mathbf{argmax}}\langle \boldsymbol{w}_{k_1}^1, \boldsymbol{h} \rangle + b_{k_1}^1, \ldots, \underset{k_T}{\mathbf{argmax}}\langle \boldsymbol{w}_{k_T}^T, \boldsymbol{h} \rangle + b_{k_T}^T)$$

$$\to \underset{k_1\ldots,k_T}{\mathbf{argmin}}\|\boldsymbol{h} - \boldsymbol{u}^{k_1\ldots,k_T}\|_2$$

Similar to Appendix B.1, we prove that the optimal solution of the loss function must satisfy the aforementioned constraints. Next, we analyze the loss function in Problem (3).

$$\sum_{t=1}^{T}\sum_{k=1}^{K_t}\sum_{i=1}^{\frac{N}{K_t}}c^t\mathbf{log}(1+\sum_{l=1,l\neq k}^{K_t}\mathbf{exp}(\boldsymbol{w}_l^t\boldsymbol{h}_{k,i}^t-\boldsymbol{w}_k^t\boldsymbol{h}_{k,i}^t+b_l^t-b_k^t))+\lambda_{\boldsymbol{H}}\|\boldsymbol{H}\|_F^2+\lambda_{\boldsymbol{W}}\sum_{t=1}^{T}c^t\|\boldsymbol{W}^t\|_F^2+\lambda_{\boldsymbol{b}}\sum_{t=1}^{T}c^t\|\boldsymbol{b}^t\|_2^2$$

$$\overset{(f)}{\geq}\sum_{t=1}^{T}\sum_{k=1}^{K_t}\sum_{i=1}^{\frac{N}{K_t}}c^t\mathbf{log}(1+(K_t-1)\mathbf{exp}(-\frac{K_t}{K_t-1}(\boldsymbol{w}_k^t\boldsymbol{h}_{k,i}^t+b_k^t)))$$
$$+\lambda_{\boldsymbol{H}}\|\boldsymbol{H}\|_F^2+\lambda_{\boldsymbol{W}}\sum_{t=1}^{T}c^t\|\boldsymbol{W}^t\|_F^2+\lambda_{\boldsymbol{b}}\sum_{t=1}^{T}c^t\|\boldsymbol{b}^t\|_2^2 \qquad (f)$$

$$\overset{(g)}{\geq}\sum_{t=1}^{T}\sum_{k=1}^{K_t}c^t\frac{N}{K_t}\mathbf{log}(1+(K_t-1)\mathbf{exp}(-\frac{K_t^2}{(K_t-1)N}(\boldsymbol{w}_k^t\sum_{i=1}^{\frac{N}{K_t}}\boldsymbol{h}_{k,i}^t+\frac{N}{K_t}b_k^t)))$$
$$+\lambda_{\boldsymbol{H}}\|\boldsymbol{H}\|_F^2+\lambda_{\boldsymbol{W}}\sum_{t=1}^{T}c^t\|\boldsymbol{W}^t\|_F^2+\lambda_{\boldsymbol{b}}\sum_{t=1}^{T}c^t\|\boldsymbol{b}^t\|_2^2 \qquad (g)$$

$$\overset{(h)}{\geq}\sum_{t=1}^{T}c^tN\mathbf{log}(1+(K_t-1)\mathbf{exp}(-\frac{K_t}{(K_t-1)N}\sum_{k=1}^{K_t}\boldsymbol{w}_k^t\sum_{i=1}^{\frac{N}{K_t}}\boldsymbol{h}_{k,i}^t))+\lambda_{\boldsymbol{H}}\|\boldsymbol{H}\|_F^2+\lambda_{\boldsymbol{W}}\sum_{t=1}^{T}c^t\|\boldsymbol{W}^t\|_F^2$$
$$(h)$$

$$\overset{(i)}{\geq}\sum_{t=1}^{T}c^tN\mathbf{log}(1+(K_t-1)\mathbf{exp}(-\frac{K_t}{(K_t-1)N}\sum_{k=1}^{K_t}\boldsymbol{w}_k^t\sum_{i=1}^{\frac{N}{K_t}}\boldsymbol{h}_{k,i}^t))+\sqrt{\frac{4\lambda_{\boldsymbol{H}}\lambda_{\boldsymbol{W}}}{N}}\sum_{t=1}^{T}\sqrt{c^tK_t}\sum_{k=1}^{K_t}\boldsymbol{w}_k^t\sum_{i=1}^{\frac{N}{K_t}}\boldsymbol{h}_{k,i}^t$$
$$(i)$$

$$=\sum_{t=1}^{T}(c^tN\mathbf{log}(1+(K_t-1)\mathbf{exp}(-\frac{K_t}{(K_t-1)N}\sum_{k=1}^{K_t}\boldsymbol{w}_k^t\sum_{i=1}^{\frac{N}{K_t}}\boldsymbol{h}_{k,i}^t))+\sqrt{\frac{4\lambda_{\boldsymbol{H}}\lambda_{\boldsymbol{W}}c^tK_t}{N}}\sum_{k=1}^{K_t}\boldsymbol{w}_k^t\sum_{i=1}^{\frac{N}{K_t}}\boldsymbol{h}_{k,i}^t)$$

Invoking Lemma A.3, when $\lambda_{\boldsymbol{H}}\lambda_{\boldsymbol{W}}<\min_t\frac{Nc^t}{4K_t}$, the minimizer must obey $x^*>0$. Inequalities (f), (g), (h) are based on the Lemma A.2, and inequality (i) is based on:

$$\sum_{t=1}^{T}\sqrt{c^tK_t}\sum_{k=1}^{K_t}\boldsymbol{w}_k^t\sum_{i=1}^{\frac{N}{K_t}}\boldsymbol{h}_{k,i}^t=\sum_{k_1=1}^{K_1}\cdots\sum_{k_T=1}^{K_T}\sum_{i=1}^{n}\boldsymbol{h}_i^{k_1,\ldots,k_T}\sum_{j=1}^{T}\sqrt{c^tK_t}\boldsymbol{w}_{k_j}^j$$
$$\leq\sum_{k_1=1}^{K_1}\cdots\sum_{k_T=1}^{K_T}\sum_{i=1}^{n}(\lambda\|\boldsymbol{h}_i^{k_1,\ldots,k_T}\|_2^2+\frac{1}{4\lambda}\|\sum_{j=1}^{T}\sqrt{c^tK_t}\boldsymbol{w}_{k_j}^j\|_2^2)$$
$$=\sum_{k_1=1}^{K_1}\cdots\sum_{k_T=1}^{K_T}\sum_{i=1}^{n}(\lambda\|\boldsymbol{h}_i^{k_1,\ldots,k_T}\|_2^2+\frac{1}{4\lambda}\sum_{j=1}^{T}c^tK_t\|\boldsymbol{w}_{k_j}^j\|_2^2) \qquad (16)$$
$$=\lambda\|\boldsymbol{H}\|_F^2+\frac{N}{4\lambda}\sum_{t=1}^{T}c^t\|W^t\|_F^2$$
$$=\sqrt{\frac{N}{4\lambda_{\boldsymbol{H}}\lambda_{\boldsymbol{W}}}}(\lambda_{\boldsymbol{H}}\|\boldsymbol{H}\|_F^2+\lambda_{\boldsymbol{W}}\sum_{i=1}^{T}c^t\|\boldsymbol{W}\|_F^2).$$

where $\lambda=\sqrt{\frac{N\lambda_{\boldsymbol{H}}}{4\lambda_{\boldsymbol{W}}}}$ and the condition for equality is:

$$\boldsymbol{h}_i^{k_1,\ldots,k_T}=\frac{1}{2\lambda}\sum_{j=1}^{T}\sqrt{c^tK_t}\boldsymbol{w}_{k_j}^j \qquad (17)$$

When one of the conditions is not met, the loss will be strictly greater than the minimum value. Combining all the conditions for equality, we can conclude that the optimal solution must satisfy the general SSMTC-NC.

### B.3 Proof of Theorem 3.2

The loss in Problem (4) can be viewed as a weighted sum of the entropy losses of multiple independent variables $(\boldsymbol{W}^t, \boldsymbol{H}^t)$. Thus, we decompose the loss into a weighted sum of multiple entropy losses and prove that for each loss, the minimizers $(\boldsymbol{W}^t, \boldsymbol{H}^t)$ must satisfy Neural Collapse.

$$
\begin{aligned}
& g(\boldsymbol{W}^1, \ldots, \boldsymbol{W}^T, \boldsymbol{H}^1, \ldots, \boldsymbol{H}^T, \boldsymbol{b}^1, \ldots, \boldsymbol{b}^T) \\
& = \sum_{t=1}^{T} c^t \sum_{k=1}^{K_t} \sum_{i=1}^{n_t} \log\left(1 + \sum_{l=1, l \neq k}^{K_t} \exp(\boldsymbol{w}_l^t \boldsymbol{h}_{k,i}^t - \boldsymbol{w}_k^t \boldsymbol{h}_{k,i}^t + b_l^t - b_k^t)\right) \\
& \quad + \sum_{t=1}^{T} c^t (\lambda_{\boldsymbol{H}} \|\boldsymbol{H}^t\|_F^2 + \lambda_{\boldsymbol{W}} \|\boldsymbol{W}^t\|_F^2 + \lambda_{\boldsymbol{b}} \|\boldsymbol{b}^t\|_2^2)
\end{aligned}
\tag{18}
$$

The specific form of the loss is shown in (18), which, as in equation (19), is decomposed into task-specific loss $g^t$.

$$
\begin{aligned}
& g(\boldsymbol{W}^1, \ldots, \boldsymbol{W}^T, \boldsymbol{H}^1, \ldots, \boldsymbol{H}^T, \boldsymbol{b}^1, \ldots, \boldsymbol{b}^T) = \sum_{t=1}^{T} c^t g^t(\boldsymbol{W}^t, \boldsymbol{H}^t, \boldsymbol{b}^t) \\
& g^t(\boldsymbol{W}^t, \boldsymbol{H}^t, \boldsymbol{b}^t) = \sum_{k=1}^{K_t} \sum_{i=1}^{n_t} \log\left(1 + \sum_{l=1, l \neq k}^{K} \exp(\boldsymbol{w}_l^t \boldsymbol{h}_{k,i}^t - \boldsymbol{w}_k^t \boldsymbol{h}_{k,i}^t + b_l^t - b_k^t)\right) \\
& \quad\quad\quad\quad\quad\quad\quad\quad + \lambda_{\boldsymbol{H}} \|\boldsymbol{H}^t\|_F^2 + \lambda_{\boldsymbol{W}} \|\boldsymbol{W}^t\|_F^2 + \lambda_{\boldsymbol{b}} \|\boldsymbol{b}^t\|_2^2
\end{aligned}
\tag{19}
$$

Similar to the proof of Theorem 3.1, since subtracting the mean from each $\boldsymbol{w}_k^t$ and $b_k^t$ leaves the cross-entropy loss unchanged while reducing the regularization loss, the optimal solution must satisfy the following condition:

$$
\begin{aligned}
\sum_{k=1}^{K_t} \boldsymbol{w}_k^t &= \boldsymbol{0}, \forall t \in [T] \\
\sum_{k=1}^{K_t} b_k^t &= 0, \forall t \in [T]
\end{aligned}
\tag{20}
$$

Observe that:

$$g^t = \sum_{k=1}^{K_t} \sum_{i=1}^{n_t} \log(1 + \sum_{l=1,l\neq k}^{K_t} \exp(\boldsymbol{w}_l^t \boldsymbol{h}_{k,i}^t - \boldsymbol{w}_k^t \boldsymbol{h}_{k,i}^t + b_l^t - b_k^t)) + \lambda_{\boldsymbol{H}} \|\boldsymbol{H}^t\|_F^2 + \lambda_{\boldsymbol{W}} \|\boldsymbol{W}^t\|_F^2 + \lambda_{\boldsymbol{b}} \|\boldsymbol{b}^t\|_2^2$$

$$\overset{(j)}{\geq} \sum_{k=1}^{K_t} \sum_{i=1}^{n_t} \log(1 + (K_t - 1)\exp(-\frac{K_t}{K_t - 1}(\boldsymbol{w}_k^t \boldsymbol{h}_{k,i}^t + b_k^t))) + \lambda_{\boldsymbol{H}} \|\boldsymbol{H}^t\|_F^2 + \lambda_{\boldsymbol{W}} \|\boldsymbol{W}^t\|_F^2 + \lambda_{\boldsymbol{b}} \|\boldsymbol{b}^t\|_2^2$$

(j)

$$\overset{(k)}{\geq} \sum_{k=1}^{K_t} n_t \log(1 + (K_t - 1)\exp(-\frac{K_t}{n_t(K_t - 1)}(\boldsymbol{w}_k^t \sum_{i=1}^{n_t} \boldsymbol{h}_{k,i}^t + n_t b_k^t))) + \lambda_{\boldsymbol{H}} \|\boldsymbol{H}^t\|_F^2 + \lambda_{\boldsymbol{W}} \|\boldsymbol{W}^t\|_F^2 + \lambda_{\boldsymbol{b}} \|\boldsymbol{b}^t\|_2^2$$

(k)

$$\overset{(l)}{\geq} K_t n_t \log(1 + (K_t - 1)\exp(-\frac{1}{n_t(K_t - 1)} \sum_{k=1}^{K_t} \boldsymbol{w}_k^t \sum_{i=1}^{n_t} \boldsymbol{h}_{k,i}^t)) + \lambda_{\boldsymbol{H}} \|\boldsymbol{H}^t\|_F^2 + \lambda_{\boldsymbol{W}} \|\boldsymbol{W}^t\|_F^2$$

(l)

$$\overset{(m)}{\geq} K_t n_t \log(1 + (K_t - 1)\exp(-\frac{1}{n_t(K_t - 1)} \sqrt{\frac{n_t}{4\lambda_{\boldsymbol{H}}\lambda_{\boldsymbol{W}}}}(\lambda_{\boldsymbol{H}} \|\boldsymbol{H}^t\|_F^2 + \lambda_{\boldsymbol{W}} \|\boldsymbol{W}^t\|_F^2))) + \lambda_{\boldsymbol{H}} \|\boldsymbol{H}^t\|_F^2 + \lambda_{\boldsymbol{W}} \|\boldsymbol{W}^t\|_F^2$$

(m)

Invoking Lemma A.3 with $\alpha = K_t n_t$, $\beta = K_t - 1$, $\gamma = \frac{1}{n_t(K_t-1)} \sqrt{\frac{n_t}{4\lambda_{\boldsymbol{H}}\lambda_{\boldsymbol{W}}}}$ and $x = \lambda_{\boldsymbol{H}} \|\boldsymbol{H}^t\|_F^2 + \lambda_{\boldsymbol{W}} \sum_{t=1}^T \|\boldsymbol{W}^t\|_F^2$, we get that if $\lambda_{\boldsymbol{H}}\lambda_{\boldsymbol{W}} \geq \frac{n_t}{4}$, the minimizer is $(\boldsymbol{W}^t, \boldsymbol{H}^t) = (\boldsymbol{0}, \boldsymbol{0})$ and otherwise, the minimizer must obey $x^* > 0$. In inequalities (j), (k), and (l). we use Lemma A.2, the conditions under which equality holds are as follows:

$$\begin{aligned}
&\forall l_1, l_2 \in [K_t] \setminus \{k\}, \boldsymbol{w}_{l_1}^t \boldsymbol{h}_{k,i}^t + b_{l_1}^t = \boldsymbol{w}_{l_2}^t \boldsymbol{h}_{k,i}^t + b_{l_2}^t \\
&\forall i_1, i_2 \in [n_t], \boldsymbol{w}_k^t \boldsymbol{h}_{k,i_1}^t = \boldsymbol{w}_k^t \boldsymbol{h}_{k,i_2}^t \\
&\forall k_1, k_2 \in [K_t], \boldsymbol{w}_{k_1}^t \sum_{i=1}^{n_t} \boldsymbol{h}_{k_1,i}^t + n_t b_{k_1}^t = \boldsymbol{w}_{k_2}^t \sum_{i=1}^{n_t} \boldsymbol{h}_{k_2,i}^t + n_t b_{k_2}^t \\
&\forall k \in [K], b_k^t = 0
\end{aligned}$$

(21)

Inequality (m) is based on:

$$\begin{aligned}
\boldsymbol{w}_k^t \sum_{i=1}^{n^t} \boldsymbol{h}_{k,i}^t &\leq \lambda \|\boldsymbol{w}_k^t\|_2^2 + \frac{1}{4\lambda} \|\sum_{i=1}^{n_t} \boldsymbol{h}_{k,i}^t\|_2^2 \\
&\leq \lambda \|\boldsymbol{w}_k^t\|_2^2 + \frac{n_t}{4\lambda} \sum_{i=1}^{n_t} \|\boldsymbol{h}_{k,i}^t\|_2^2
\end{aligned}$$

(22)

where $\lambda = \sqrt{\frac{n_t \lambda_{\boldsymbol{W}}}{4\lambda_{\boldsymbol{H}}}}$ and the condition for equality to hold is:

$$\forall i \in [n_t], \boldsymbol{w}_k^t = \sqrt{\frac{\lambda_{\boldsymbol{H}} n_t}{\lambda_{\boldsymbol{W}}}} \boldsymbol{h}_{k,i}^t$$

(23)

Based on the above condition, we conclude that the classifier and features must satisfy the NC conditions within each task.

## B.4 SSMTC-NC IN THE TASK-BALANCED CASE

**Theorem B.1.** *Consider two binary classification tasks, the training data is task-balanced, and the number of samples with label $(k_1, k_2)$ is $n^{k_1, k_2}(k_1 \in [2], k_2 \in [2])$. Assume that the number of all samples is $N$, then any global minimizers of Problem 3 satisfies the following conditions: Consider two binary classification tasks, the training data is task-balanced, and the number of samples with label $(k_1, k_2)$ is $n^{k_1, k_2}(k_1 \in [2], k_2 \in [2])$. Assume that the number of all samples is $N$ and $\lambda_{\boldsymbol{H}}\lambda_{\boldsymbol{W}} < \frac{n^{k_1, k_2}}{2}$, then any global minimizers of Problem (3) satisfies the following conditions:*

1. $\boldsymbol{h}_j^{k_1,k_2} = \boldsymbol{\mu}^{k_1,k_2}$

2. $\|\boldsymbol{w}_k^t\|_2 = \|\boldsymbol{w}_{k'}^t\|_2, \langle \tilde{\boldsymbol{w}}_k^t, \tilde{\boldsymbol{w}}_{k'}^t \rangle = 2\delta_{k,k'} - 1, \forall k, k' \in [2], t \in [2]$

3. As $n^{k_1,k_2}$ increases, $\cos(\boldsymbol{w}_{k_1}^1, \boldsymbol{w}_{k_2}^2)$ increases.

4. $\tilde{\boldsymbol{h}}_j^{k_1,k_2} = \dfrac{\boldsymbol{w}_{k_1}^1 + \boldsymbol{w}_{k_2}^2}{\|\boldsymbol{w}_{k_1}^1 + \boldsymbol{w}_{k_2}^2\|_2}$

5. $\left( \underset{k_1}{\mathbf{argmax}} \langle \boldsymbol{w}_{k_1}^1, \boldsymbol{h} \rangle + b_{k_1}^1, \underset{k_2}{\mathbf{argmax}} \langle \boldsymbol{w}_{k_2}^2, \boldsymbol{h} \rangle + b_{k_2}^2 \right) = \underset{k_1,k_2}{\mathbf{argmin}} \|\boldsymbol{h} - \boldsymbol{u}^{k_1,k_2}\|_2$

*Proof.* The key idea of the proof of Theorem 5.1 is to lower bound the entropy loss in Problem (3), and obtain the form of $\cos(\boldsymbol{w}_{k_1}^1, \boldsymbol{w}_{k_2}^2)(k_1 \in [2], k_2 \in [2])$. We denote the cross-entropy loss for samples with labels $(k_1, k_2, \ldots, k_T)$ as $u(k_1, k_2, \ldots, k_T)$. Different from the theories in Appendix B.1, we can only scale the cross-entropy loss for samples with the same labels.

$$
\begin{aligned}
&u(k_1, k_2) \\
&= \sum_{t=1}^{2} \sum_{i=1}^{n^{k_1,k_2}} \log(1 + \sum_{l \neq k_t}^{2} \exp(\boldsymbol{w}_l^t \boldsymbol{h}_i^{k_1,k_2} - \boldsymbol{w}_{k_t}^t \boldsymbol{h}_i^{k_1,k_2} + b_l^t - b_{k_t}^t)) \\
&= \sum_{t=1}^{2} \sum_{i=1}^{n^{k_1,k_2}} \log(1 + \exp(-2(\boldsymbol{w}_{k_t}^t \boldsymbol{h}_i^{k_1,k_2} + b_{k_t}^t))) \\
&\geq \sum_{t=1}^{T} n^{k_1,k_2} \log(1 + (K-1)\exp(-2(\boldsymbol{w}_{k_t}^t \frac{1}{n^{k_1,k_2}} \sum_{i=1}^{n^{k_1,k_2}} \boldsymbol{h}_i^{k_1,k_2} + b_{k_t}^t))) \\
&\geq 2n^{k_1,k_2} \log(1 + \exp(-\sum_{t=1}^{2}(\boldsymbol{w}_{k_t}^t \frac{1}{n^{k_1,k_2}} \sum_{i=1}^{n^{k_1,k_2}} \boldsymbol{h}_i^{k_1,k_2} + b_{k_t}^t)))
\end{aligned}
\tag{24}
$$

In 24, the second equality is due to the condition 10. Combining the sum of all the cross-entropy loss and the regularization term, we derive the following composite loss function:

$$
\begin{aligned}
&g(\boldsymbol{W}^1, \boldsymbol{W}^2, \boldsymbol{H}, \boldsymbol{b}^1, \boldsymbol{b}^2) \\
&= \sum_{k_1=1}^{2} \sum_{k_2=1}^{2} u(k_1, k_2) + \lambda_{\boldsymbol{H}} \|\boldsymbol{H}\|_F^2 + \lambda_{\boldsymbol{W}} \sum_{t=1}^{2} \|\boldsymbol{W}^t\|_F^2 + \lambda_{\boldsymbol{b}} \sum_{t=1}^{2} \|\boldsymbol{b}^t\|_2^2 \\
&\geq \sum_{k_1=1}^{2} \sum_{k_2=1}^{2} (u(k_1, k_2) + \lambda_{\boldsymbol{H}} \sum_{i=1}^{n^{k_1,k_2}} \|\boldsymbol{h}_i^{k_1,k_2}\|_2^2 + \frac{\lambda_{\boldsymbol{W}}}{2} \|\sum_{t=1}^{2} \boldsymbol{w}_{k_t}^t\|_2^2) \\
&\geq \sum_{k_1=1}^{2} \sum_{k_2=1}^{2} (2n^{k_1,k_2} \log(1 + \exp(-\frac{1}{n^{k_1,k_2}} \sqrt{\frac{n^{k_1,k_2}}{2\lambda_{\boldsymbol{W}} \lambda_{\boldsymbol{H}}}} \\
&(\lambda_{\boldsymbol{H}} \sum_{i=1}^{n^{k_1,k_2}} \|\boldsymbol{h}_i^{k_1,k_2}\|_2^2 + \frac{\lambda_{\boldsymbol{W}}}{2} \|\sum_{t=1}^{2} \boldsymbol{w}_{k_t}^t\|_2^2)) + \lambda_{\boldsymbol{H}} \sum_{i=1}^{n^{k_1,k_2}} \|\boldsymbol{h}_i^{k_1,k_2}\|_2^2 + \frac{\lambda_{\boldsymbol{W}}}{2} \|\sum_{t=1}^{2} \boldsymbol{w}_{k_t}^t\|_2^2)
\end{aligned}
\tag{25}
$$

We can get the conditions that need to be satisfied as follows:

$$
\boldsymbol{w}_{k_t}^t \boldsymbol{h}_i^{k_1,k_2} = \boldsymbol{w}_{k_t}^t \boldsymbol{h}_j^{k_1,k_2}, \forall i, j \in [n^{k_1,k_2}], t \in [2]
\tag{26}
$$

$$
\boldsymbol{w}_{k_1}^1 \boldsymbol{h}_i^{k_1,k_2} = \boldsymbol{w}_{k_2}^2 \boldsymbol{h}_i^{k_1,k_2}, \forall k_1, k_2 \in [2], i \in [n^{k_1,k_2}]
\tag{27}
$$

$$
\boldsymbol{b}^t = \boldsymbol{0}, \forall t \in [2]
\tag{28}
$$

$$
\sqrt{\frac{2\lambda_{\boldsymbol{H}} n^{k_1,k_2}}{\lambda_{\boldsymbol{W}}}} \boldsymbol{h}_i^{k_1,k_2} = \sum_{t=1}^{2} \boldsymbol{w}_{k_t}^t, \forall k_1, k_2 \in [2], i \in [n^{k_1,k_2}]
\tag{29}
$$

Table 2: Dataset illustration

| Setting | Dataset | T | K |
|---------|---------|---|---|
| SSMTC-NC | Multi-MNIST | 2 | $\geq 2$ |
| | Multi-CIFAR10 | 2 | $\geq 2$ |
| | CIFAR100-Cross | $\geq 2$ | $\geq 2$ |
| | CelebA | $\geq 2$ | 2 |
| MSMTC-NC | MNIST-Split | 2 | $\geq 2$ |
| | CIFAR-Split | 2 | $\geq 2$ |
| | CIFAR100-Split-5x20 | 20 | 5 |

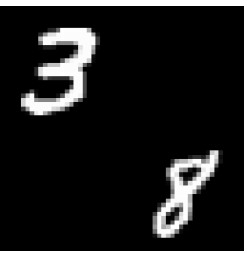 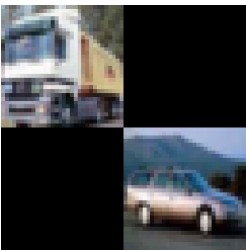

Figure 5: The image on the left shows examples from Multi-MNIST with the label (3, 8), while the image on the right shows examples from Multi-CIFAR10 with the label (truck, automobile).

Combining all the conditions metioned, we can conclude that NC1, NC2, NC4, NC5 need to be satisfied. Moreover, we conclude that:

$$\|\boldsymbol{w}\|_2^2 = \|\boldsymbol{w}_{k_1}^1\|_2^2 = \|\boldsymbol{w}_{k_2}^2\|_2^2, \forall k_1, k_2 \in [2] \tag{30}$$

$$(\sum_{t=1}^2 \boldsymbol{w}_{k_t}^t)^2 = 2\|\boldsymbol{w}\|_2^2 + 2\|\boldsymbol{w}\|_2^2 \cos(\boldsymbol{w}_{k_1}^1, \boldsymbol{w}_{k_2}^2)$$
$$= \sqrt{\frac{2\lambda_H n^{k_1,k_2}}{\lambda_W}} \log(\sqrt{\frac{2n^{k_1,k_2}}{\lambda_H \lambda_W}} - 1) \tag{31}$$

Let $\omega^{k_1,k_2} = (\sum_{t=1}^2 \boldsymbol{w}_{k_t}^t)^2$. And the $\cos(\boldsymbol{w}_{k_1}^1, \boldsymbol{w}_{k_2}^2) = \frac{2\omega^{k_1,k_2}}{\omega^{k_1,k_2} + \omega^{k_1,3-k_2}} - 1$. From the analytical expression of $\cos(\boldsymbol{w}_{k_1}^1, \boldsymbol{w}_{k_2}^2)$, we can conclude that $\cos(\boldsymbol{w}_{k_1}^1, \boldsymbol{w}_{k_2}^2)$ increases with growing $n^{k_1,k_2}$ when fixing the number of training samples $N$. So as $p_{Y_1,Y_2}(k_1, k_2) = \frac{n^{k_1,k_2}}{N}$ increases, $\cos(\boldsymbol{w}_{k_1}^1, \boldsymbol{w}_{k_2}^2)$ increases. $\qquad \square$

## C  DATASET ILLUSTRATION AND VISUALIZATION

### C.1  MULTI-MNIST AND MULTI-CIFAR10

Multi-MNIST and Multi-CIFAR10 are used in the case of two tasks in the SSMTC (Single-Source Multi-Task Classification) scenario. As illustrated in Figure 5, we follow Sener & Koltun (2018); Li et al. (2024) to place the images from any two classes in MNIST dataset in the top-left and bottom-right corners, respectively, with the empty areas filled using zero-padding. The same approach is applied to the CIFAR10 dataset. We use Multi-MNIST-P-Q to refer to the dataset constructed using the first P classes and the first Q classes of MNIST. Similarly, Multi-CIFAR10-P-Q denotes the dataset constructed using the first P classes and the first Q classes of CIFAR10. By default, Multi-MNIST and Multi-CIFAR10 refer to Multi-MNIST-10-10 and Multi-CIFAR10-10-10, respectively.

## C.2 CIFAR100-CROSS AND CIFAR100-SPLIT-5X20

CIFAR100-Cross-10x10 refers to assigning the label $(i, j)(i \in \{0, 1, 2, \cdots, 9\}, j \in \{0, 1, 2, \cdots, 9\})$ to each class in the CIFAR100 dataset. CIFAR100-Cross-10x10 involves two tasks and each task classifies a task-specific label in the label $(i, j)$. Specifically, the samples of class $K \in \{0, 1, 2, \cdots, 99\}$ in CIFAR100 correspond to the samples with the label $(\lfloor K/10 \rfloor, K\%10)$ in CIFAR100-Cross-10x10. Similarly, CIFAR100-Cross-4x4x4 involves three tasks, utilizing the first $64$ classes of CIFAR100, and CIFAR100-Cross-3x3x3x3 involves four tasks, selecting the first $81$ classes.

For CIFAR100-Split-5x20, we follow Rosenbaum et al. (2018) to treat 20 coarse labels as distinct tasks and create a multi-task dataset with $2500$ train images and $500$ test images in each task.

For the MNIST-Split-5x2 dataset (CIFAR10-Split-5x2), the first five classes and the last five classes of MNIST (CIFAR10) are used as a task-specific dataset respectively.

# D DETAILED EXPERIMENTAL RESULTS

In Appendix D.1, we provide experimental results in the validation experiments of SSMTC-NC and MSMTC-NC. In Appendix D.2, we show the validation experimental results of SSMTC-NC and MSMTC-NC under different numbers of task classes and non-uniform task weights. In Appendix D.3, we show that when using the MGDA, Uncertainty Weight, PCgrad, DWA, FAMO, FairGrad algorithms to update task weights, **NC** still exists in SSMTC and MSMTC. In Appendix D.4, we conduct experiments on ImageNet, TinyImageNet and CelebA. In Appendix D.5, we run the experiments on different learning rates and $(\lambda_H, \lambda_W)$. In Appendix D.6, we show the experimental results when the train data is task-balanced. In Appendix D.7, we provide the experimental results on efficient training utilizing NC properties in MTL. We provide the specific experimental metrics in the Appendix . We run experiments on a single NVIDIA RTX 4090 (24GB) GPU.

## D.1 EXPERIMENTAL RESULTS ON SSMTC-NC AND MSMTC-NC UNDER LABEL-BALANCED CONDITIONS

Figures 6 and 7 depict the trends of the SNC1, SNC2-1, SNC2-2, SNC3, SNC4 and SNC5 metrics on the Multi-MNIST-10-10 and Multi-CIFAR10-10-10 datasets, all converging toward zero. Figures 8, 9 show the trends on CIFAR100-Cross-10x10, CIFAR100-Cross-4x4x4 and CIFAR100-Cross-3x3x3x3 datasets. Figures 10, 11 depict the trends of the MNC1, MNC2-1, MNC2-2, MNC3, MNC4 metrics on the MNIST-Split-5x2, CIFAR10-Split-5x2 and CIFAR100-Split-5x20 datasets. The convergence of all metrics toward zero validates both the SSMTC-NC and MSMTC-NC phenomena.

## D.2 SUPPLEMENTARY EXPERIMENT FOR GENERAL SSMTC-NC AND MSMTC-NC

**General SSMTC-NC**. To verify the theory in Appendix B.3 empirically, we use Multi-MNIST-4-6, Multi-MNIST-3-9, Multi-CIFAR10-4-6 and Multi-CIFAR10-3-9 to verify our results and set the task weight ratios to be inversely proportional to the class counts. Concretely, we set task weights to 1.5 and 1 for Multi-MNIST-4-6, and set task weights to 3 and 1 for Multi-MNIST-3-9. As shown in Figure 12, SSMTC-NC generalize to cases where the number of classes and the task weights are different in different tasks.

**General MSMTC-NC**. To verify the neural collapse across different task weights and different class number, we use MNIST-Split-3-4 and MNIST-Split-3-7 datasets with uniform and non-uniform task weights respectively. As shown in Figure 13, MSMTC-NC extends to cases where different tasks have varying numbers of classes and distinct task weights.

## D.3 VALIDATION EXPERIMENTS OF NC UNDER MULTIPLE MTL LEARNING METHODS

We show the experimental results on Multi-MNIST-10-10 and Multi-CIFAR10-10-10 datasets in the setting of SSMTC, and the experimental results on CIFAR100-Split-5x20 in the MSMTC setting when using the MGDA algorithm. As shown in the Figure 14 and Figure 15, when the MGDA

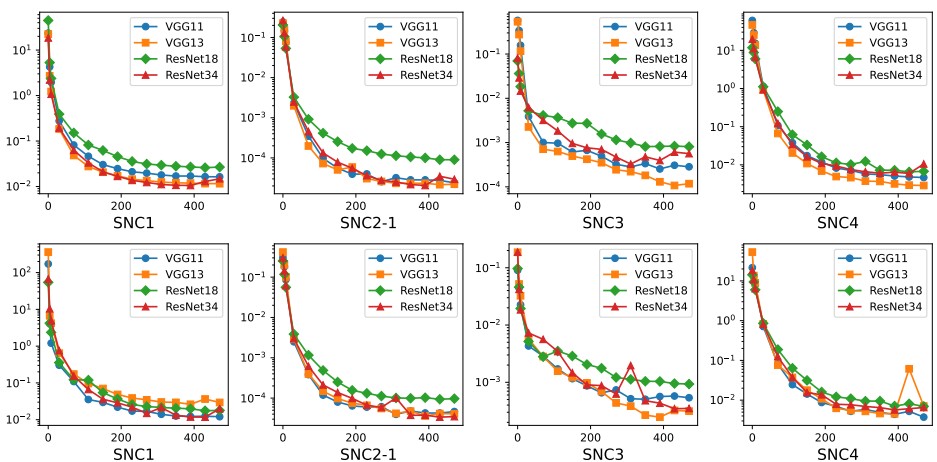

Figure 6: Illustration of SSMTC-NC across different network architectures on Multi-MNIST-10-10 (top) and Multi-CIFAR10-10-10 (bottom). The horizontal axis represents the training epochs.

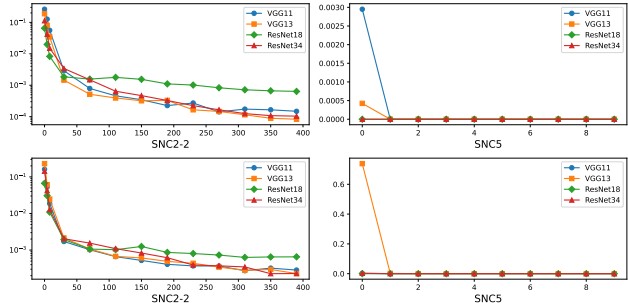

Figure 7: Illustration of SSMTC-NC across different network architectures on Multi-MNIST-10-10 (top) and Multi-CIFAR10-10-10 (bottom) (Supplementary Experimental Metrics). The horizontal axis represents the training epochs.

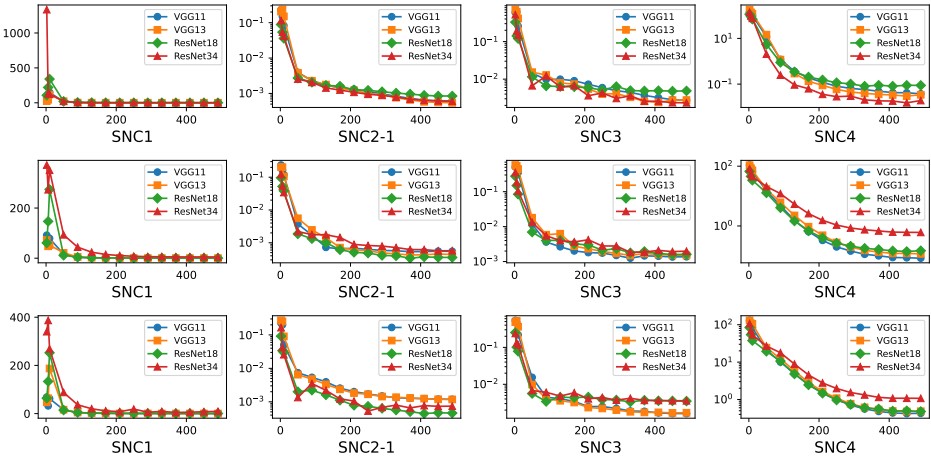

Figure 8: Illustration of SSMTC-NC across different network architectures on CIFAR100-Cross-10x10 (top), CIFAR100-Cross-4x4x4 (middle) and CIFAR100-Cross-3x3x3x3 (bottom). The horizontal axis represents the training epochs.

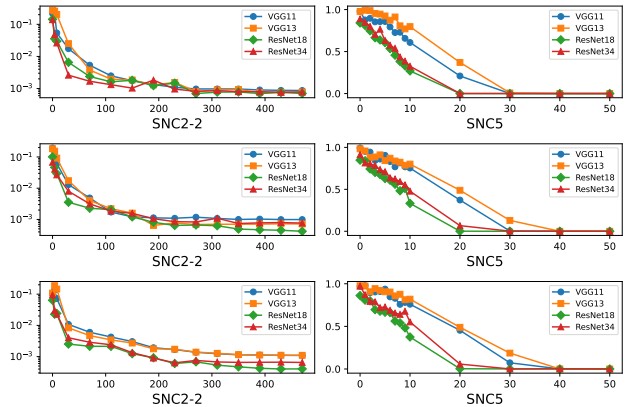

Figure 9: Illustration of SSMTC-NC across different network architectures on CIFAR100-Cross-10x10 (top), CIFAR100-Cross-4x4x4 (middle) and CIFAR100-Cross-3x3x3x3 (bottom). (Supplementary Experimental Metrics). The horizontal axis represents the training epochs.

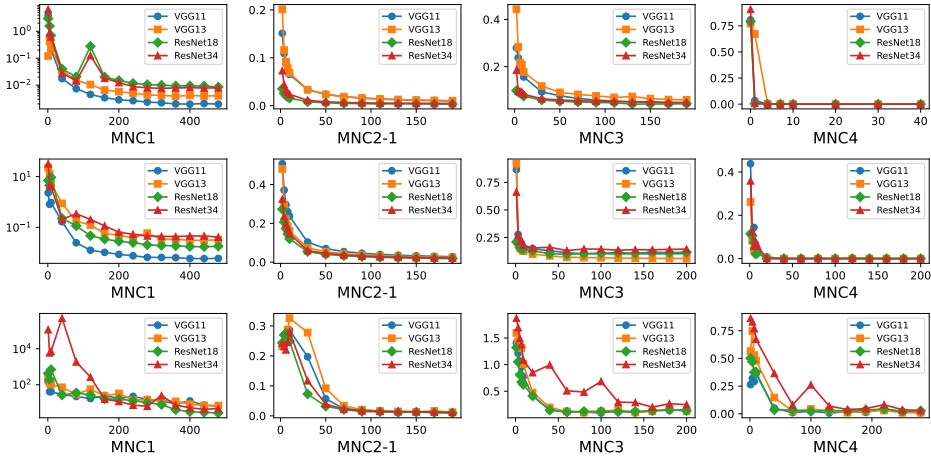

Figure 10: Illustration of MSMTC-NC across different network architectures on MNIST-Split5x2 (top), CIFAR10-Split-5x2 (middle) and CIFAR100-Split-5x20 (bottom). The horizontal axis represents the training epochs.

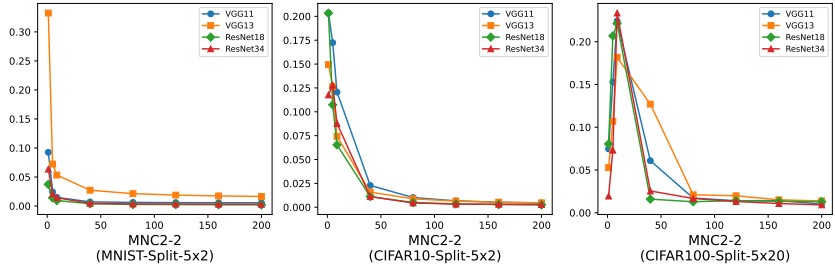

Figure 11: Illustration of MSMTC-NC (metric MNC2) across different network architectures on MNIST-Split5x2 (left), CIFAR10-Split-5x2 (middle) and CIFAR100-Split-5x20 (right). The horizontal axis represents the training epochs.

algorithm is used to update the task weights, SSMTC-NC and MSMTC-NC still exist even though

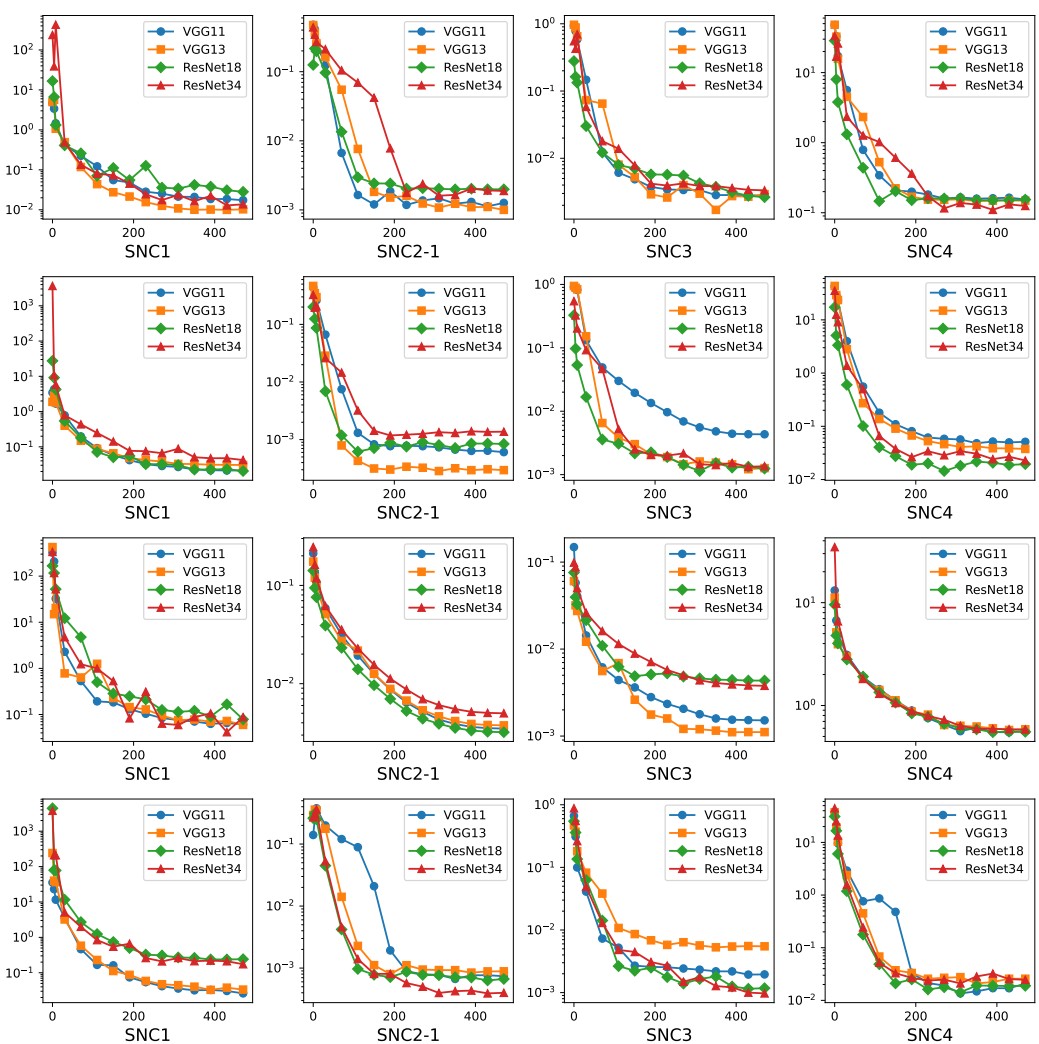

Figure 12: Illustration of SSMTC-NC across different network architectures on Multi-MNIST-3-9 (first row), Multi-MNIST-4-6 (second row), Multi-CIFAR10-3-9 (third row), and Multi-CIFAR10-4-6 (fourth row). The horizontal axis represents the training epochs.

.

the task weights are constantly changing during training, indicating that SSMTC-NC and MSMTC-NC exhibit generalization across different task weights.

Similar to MGDA experiments, we show the experimental results using Uncertainty Weight, DWA, PCgrad, FAMO, FairGrad. As shown from Figure 16 to Figure 35, SSMTC-NC and MSMTC-NC still exist when using different multiple MTL learning methods.

## D.4 EXPERIMENTS ON LARGE-SCALE DATASETS

We validate the SSMTC-NC and MSMTC-NC on ImageNet, TinyImageNet and CelebA. We utilize the first 512 classes of ImageNet to construct ImageNet-Cross-2x2x2x2x2x2x2x2x2, utilize the first 729 classes of ImageNet to construct the ImageNet-Cross-3x3x3x3x3x3, and utilize the 1000 classes of ImageNet to construct the ImageNet-Cross-10x10x10 for SSMTC. We select the 1000 classes of ImageNet to construct the ImageNet-Split-10x100 for MSMTC. We select two groups of attributes on CelebA and keep the dataset label-balanced. As shown in Figures 36, 37, 38, 39, 40, 41, 42, 43, SSMTC-NC and MSMTC-NC still exist in ImageNet, TinyImageNet and CelebA.

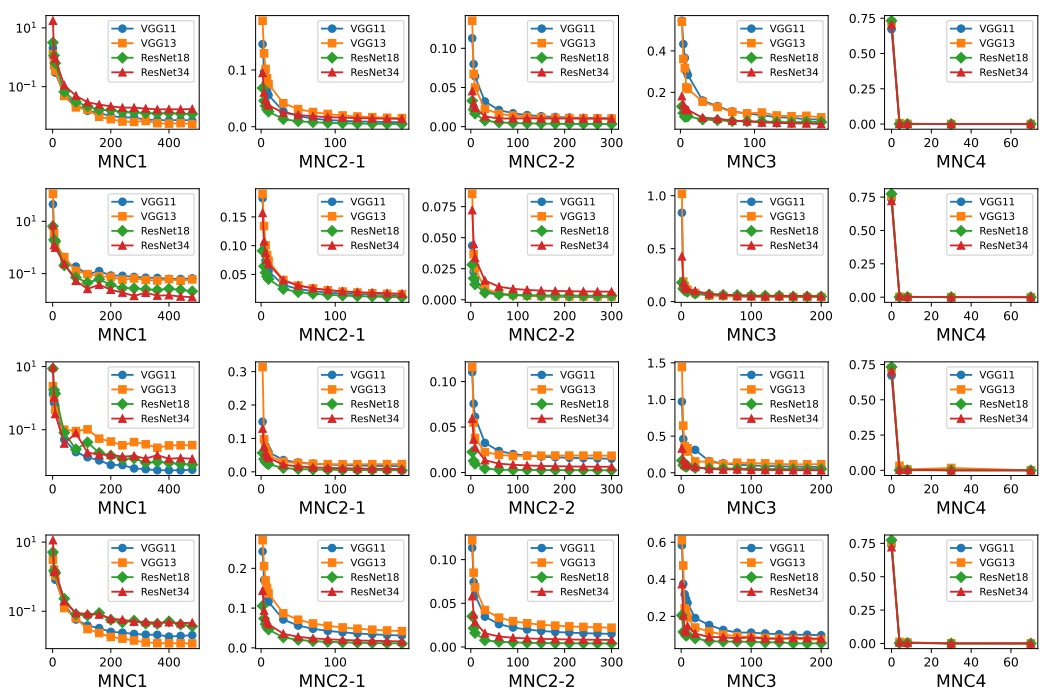

Figure 13: Illustration of MSMTC-NC across different network architectures on MNIST-Split-3-4 (first row) with uniform task weights, MNIST-Split-3-7 with uniform task weights (second row), MNIST-Split-3-4 with non-uniform task weights (third row), and MNIST-Split-3-7 with non-uniform task weights (fourth row).

.

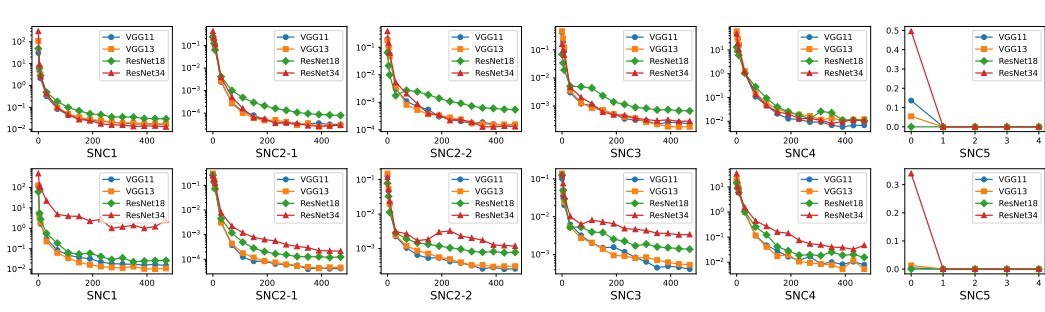

Figure 14: Illustration of SSMTC-NC across different network architectures on Multi-MNIST-10-10 (top) and Multi-CIFAR10-10-10 (bottom) by using MGDA algorithm.

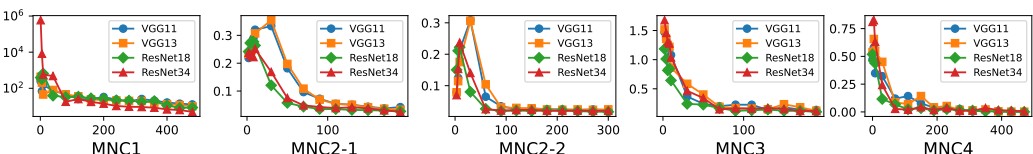

Figure 15: Illustration of MSMTC-NC on CIFAR100-Split-5x20 by using MGDA algorithm.

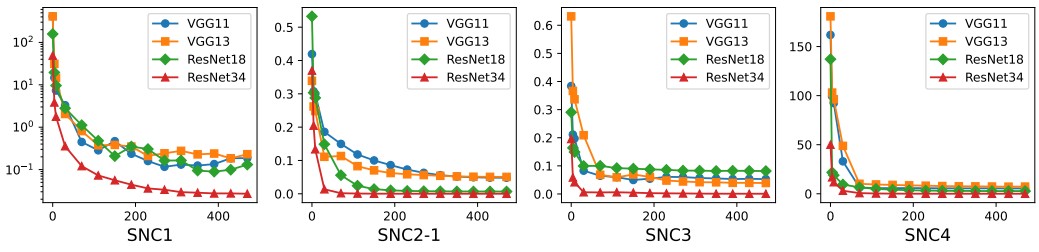

Figure 16: Experiment on Multi-MNIST using DWA method

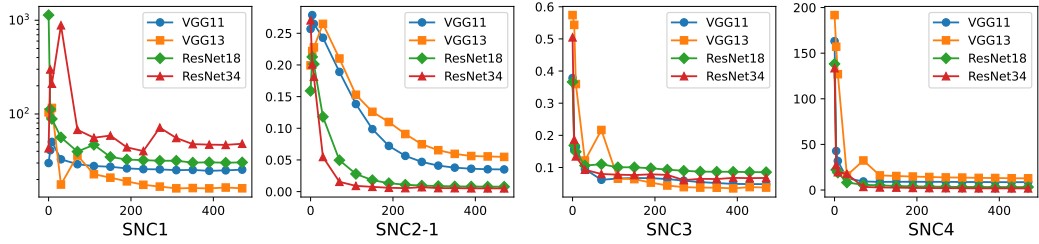

Figure 17: Experiment on Multi-CIFAR10 using DWA method

## D.5 EXPERIMENTS WITH DIFFERENT LEARNING RATES AND REGULARIZATION COEFFICIENTS

We have supplemented the experimental results for learning rates of $1 \cdot 10^{-2}$ and $1 \cdot 10^{-3}$, as well as $(\lambda_{\boldsymbol{H}}, \lambda_{\boldsymbol{W}})$ of $5 \cdot 10^{-3}$ and $5 \cdot 10^{-5}$. As shown from Figure 44 to Figure 59, the experimental results further validate the generality of our findings.

## D.6 EXPERIMENTS ON SSMTC-NC IN THE TASK-BALANCED CASE

We run experiments on the Multi-MNIST with $K = 2$, $K = 3$, $K = 10$. By varying the number of samples for each label $(k_1, k_2)$ while keeping the train data task-balanced, we find that $\cos\langle \boldsymbol{w}_{k_1}^1, \boldsymbol{w}_{k_2}^2 \rangle$ increases as the the number of samples with label $(k_1, k_2)$ increases as shown in Figures 60, 61, 62. The other NC phenomena still exist.

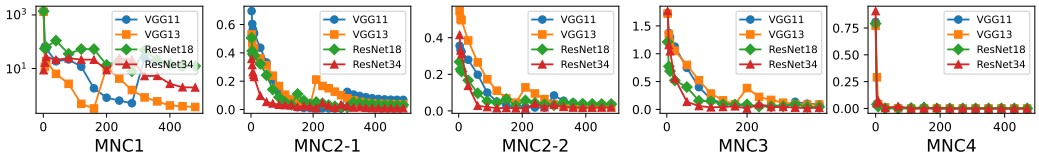

Figure 18: Experiment on MNIST-Split-5x2 using DWA method

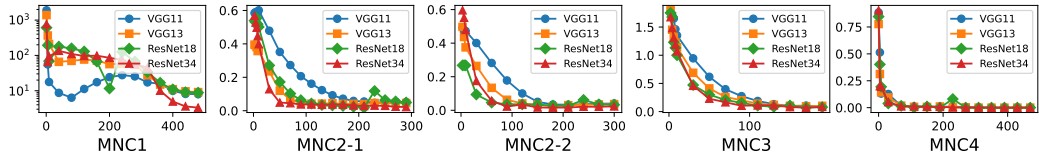

Figure 19: Experiment on CIFAR10-Split-5x2 using DWA method

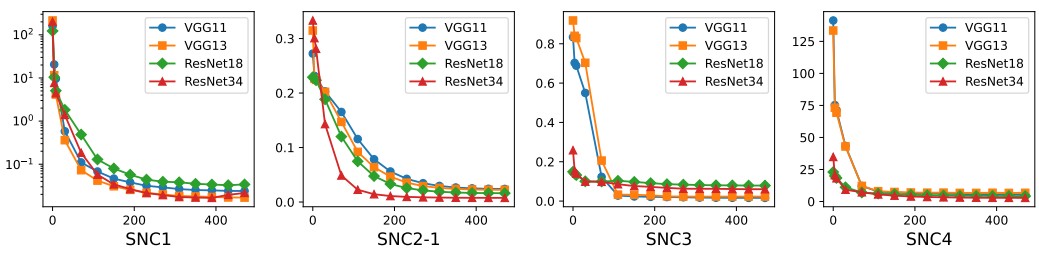

Figure 20: Experiment on Multi-MNIST using Uncertainty Weight method

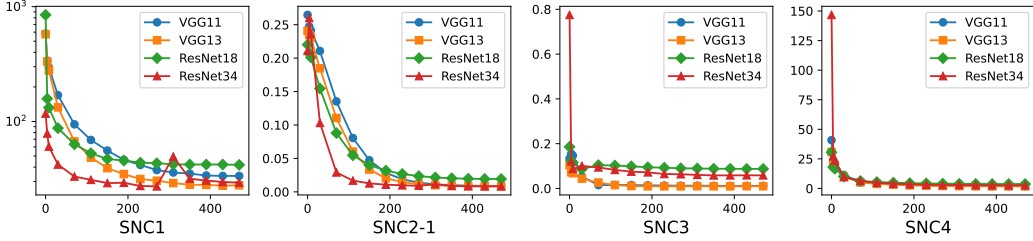

Figure 21: Experiment on Multi-CIFAR10 using Uncertainty Weight method

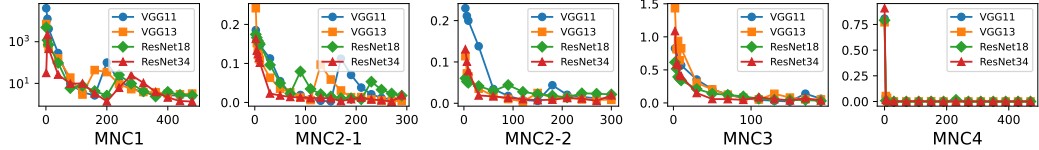

Figure 22: Experiment on MNIST-Split-5x2 using Uncertainty Weight method

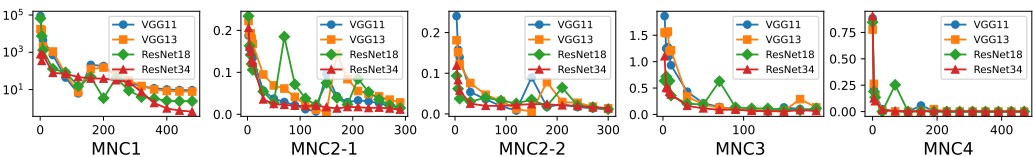

Figure 23: Experiment on CIFAR10-Split-5x2 using Uncertainty Weight method

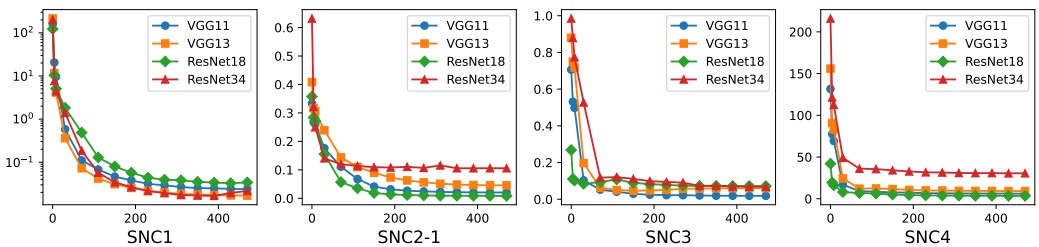

Figure 24: Experiment on Multi-MNIST using PCgrad method

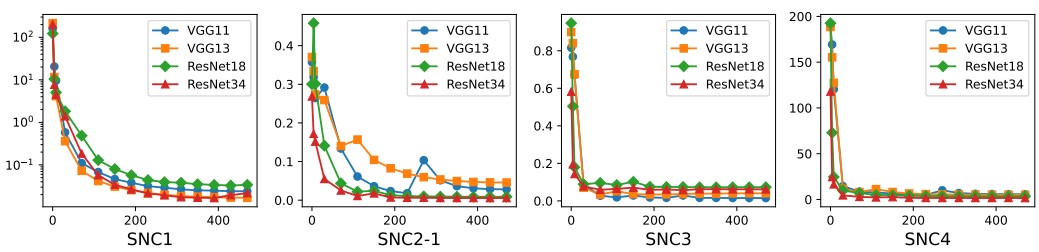

Figure 25: Experiment on Multi-CIFAR10 using PCgrad method

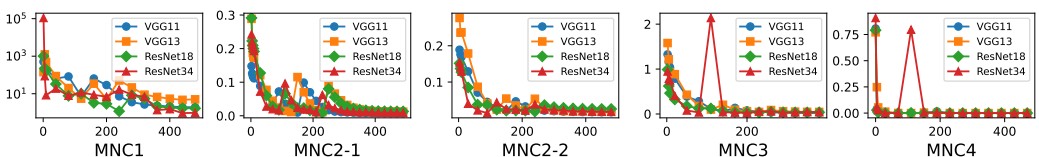

Figure 26: Experiment on MNIST-Split-5x2 using PCgrad method

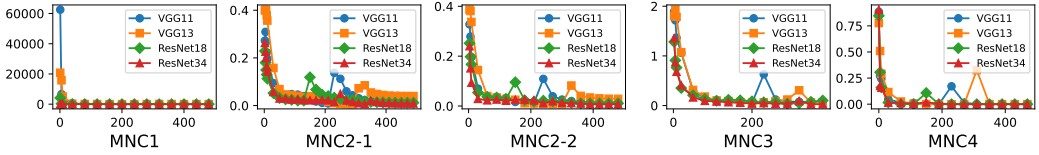

Figure 27: Experiment on CIFAR10-Split-5x2 using PCgrad method

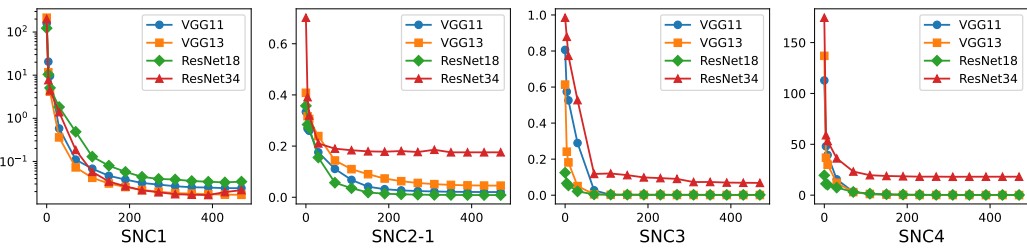

Figure 28: Experiment on Multi-MNIST using FAMO method

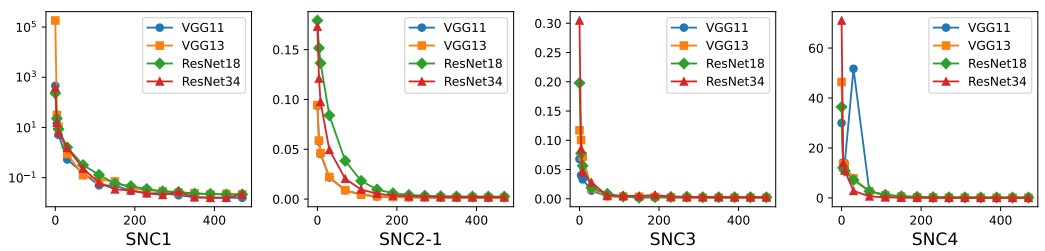

Figure 29: Experiment on Multi-CIFAR10 using FAMO method

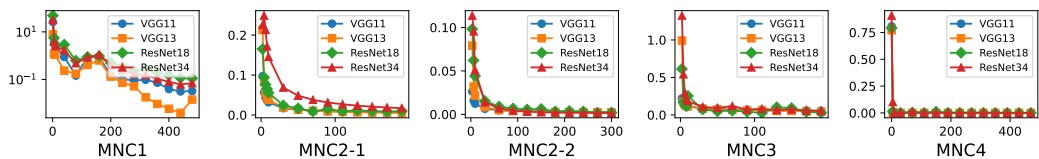

Figure 30: Experiment on MNIST-Split-5x2 using FAMO method

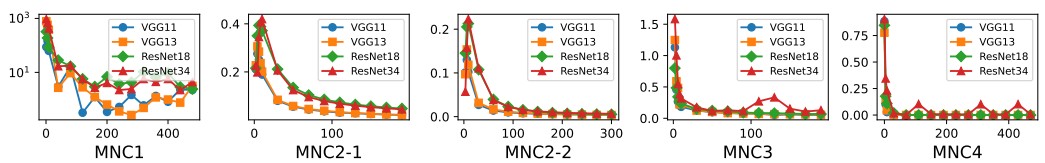

Figure 31: Experiment on CIFAR10-Split-5x2 using FAMO method

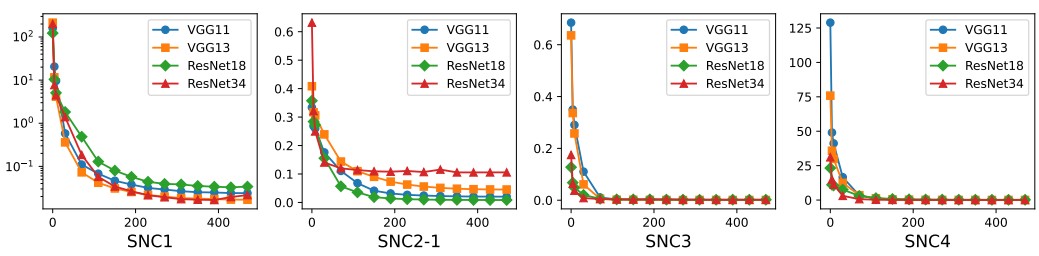

Figure 32: Experiment on Multi-MNIST using FairGrad method

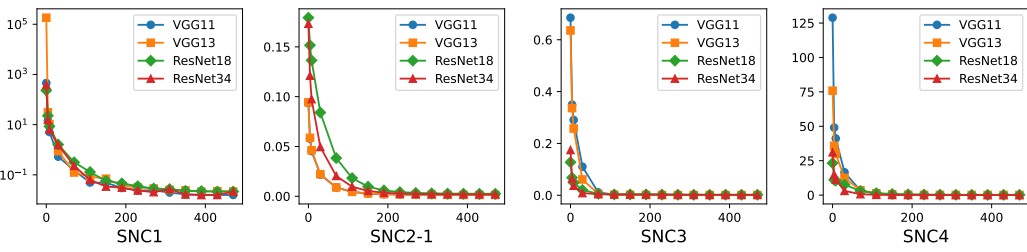

Figure 33: Experiment on Multi-CIFAR10 using FairGrad method

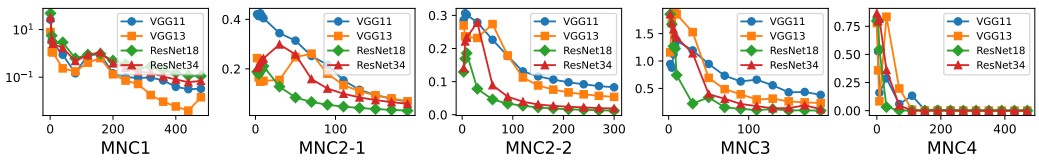

Figure 34: Experiment on MNIST-Split-5x2 using FairGrad method

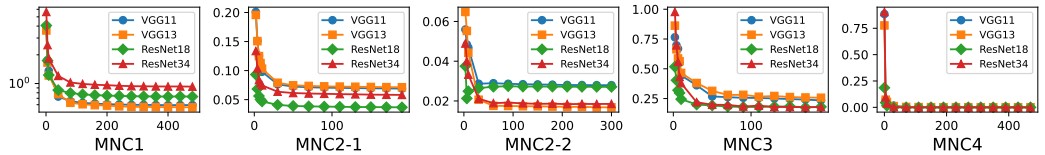

Figure 35: Experiment on CIFAR10-Split-5x2 using FairGrad method

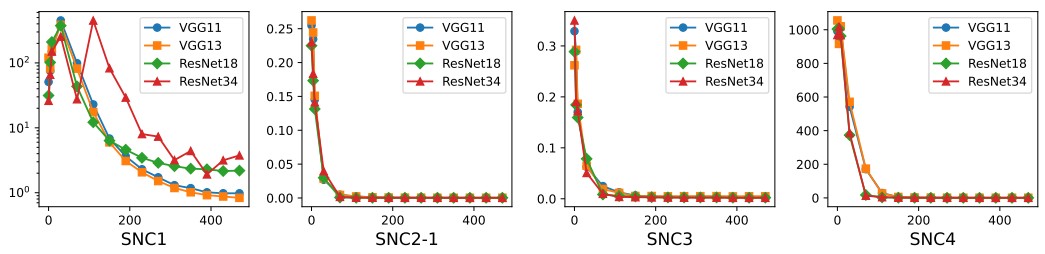

Figure 36: Experiment on ImageNet-Cross-2x2x2x2x2x2x2x2x2

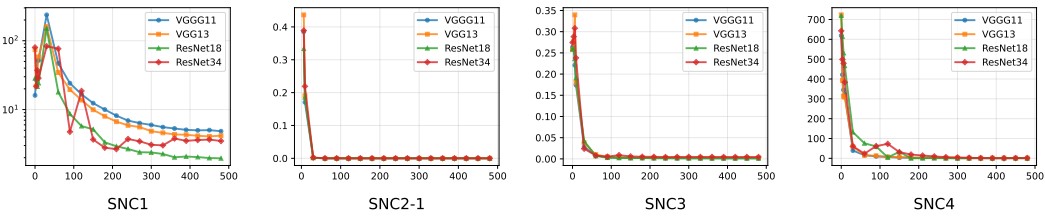

Figure 37: Experiment on ImageNet-Cross-3x3x3x3x3x3

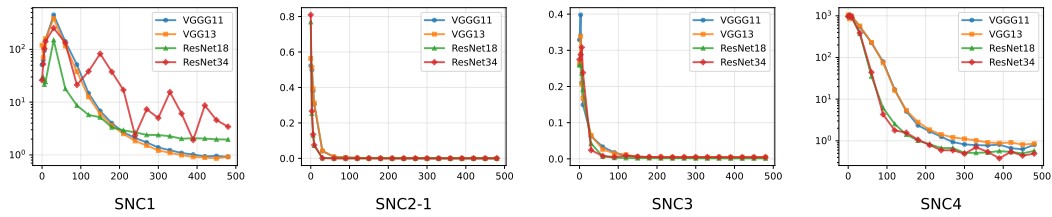

Figure 38: Experiment on ImageNet-Cross-10x10x10

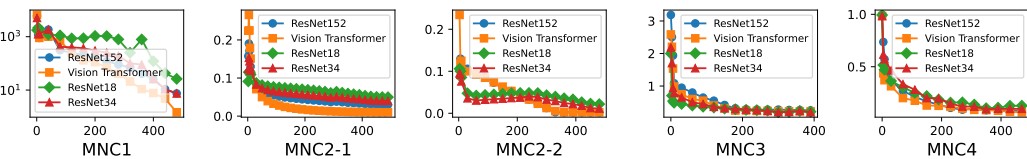

Figure 39: Experiment on ImageNet-Split-10x100

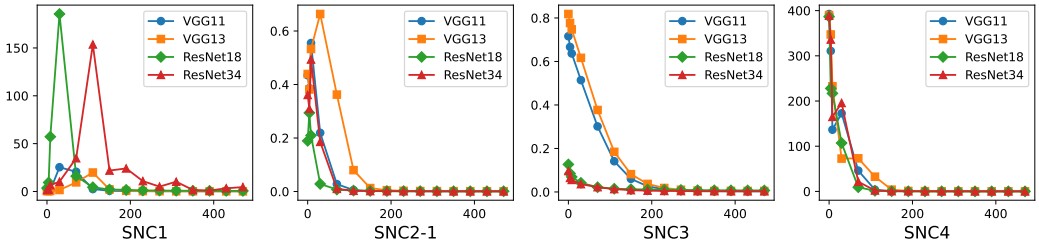

Figure 40: Experiment on TinyImageNet-Cross-10x20

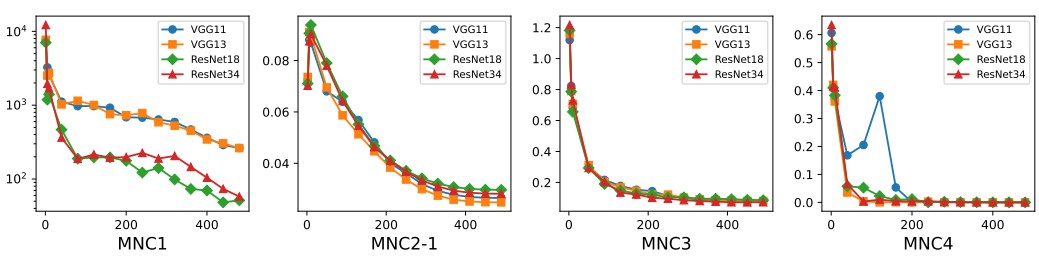

Figure 41: Experiment on TinyImageNet-Split-10x20

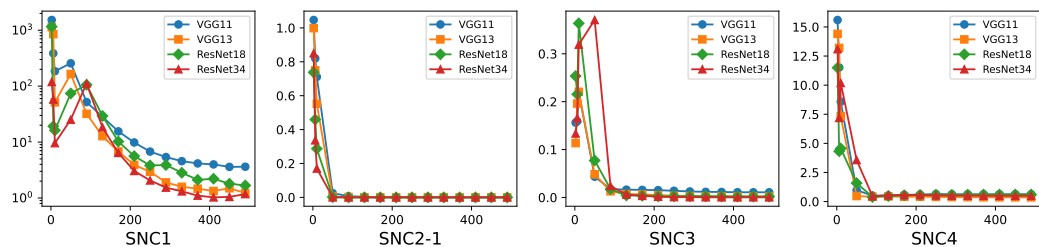

Figure 42: Experiment on CelebA on three tasks (Smiling, High_cheekbones, Mouth_Slightly_open)

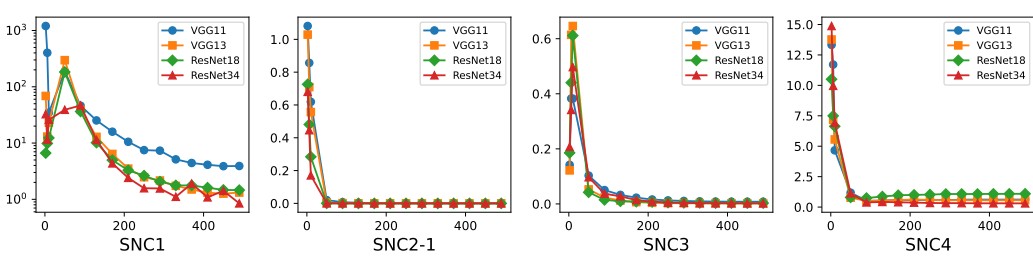

Figure 43: Experiment on CelebA on three tasks (Arched_Eyebrows, Narrow_Eyes, Bags_Under_Eyes)

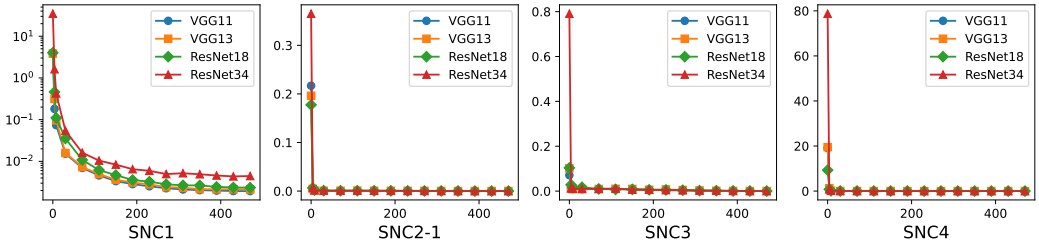

Figure 44: Experiment on Multi-MNIST using regularization terms $5 \cdot 10^{-3}$

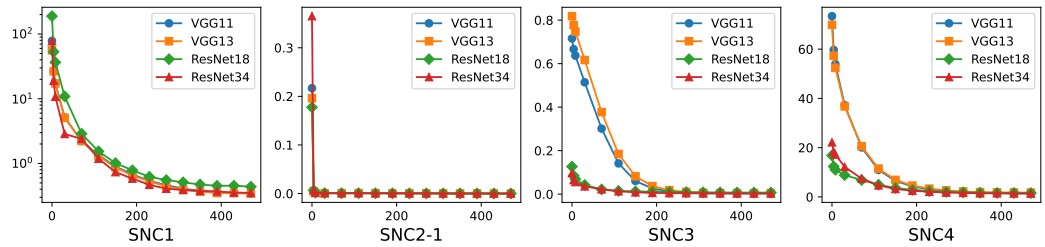

Figure 45: Experiment on Multi-MNIST using regularization terms $5 \cdot 10^{-5}$

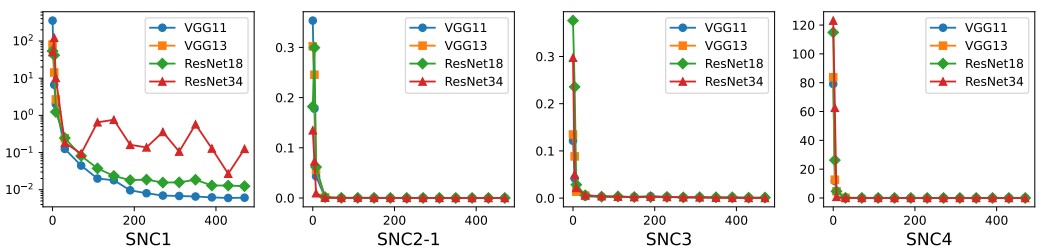

Figure 46: Experiment on Multi-CIFAR10 using regularization terms $5 \cdot 10^{-3}$

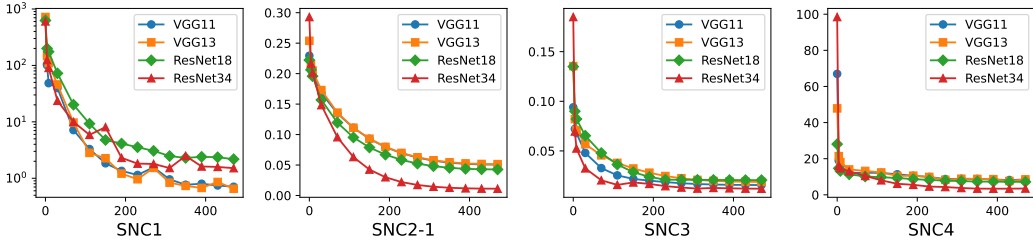

Figure 47: Experiment on Multi-CIFAR10 using regularization terms $5 \cdot 10^{-5}$

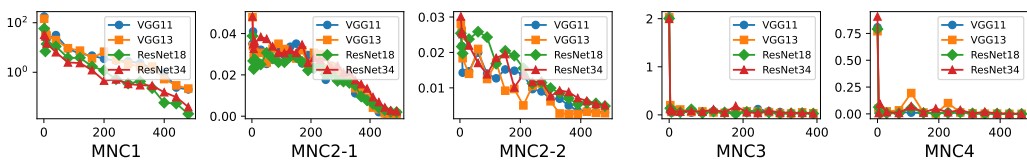

Figure 48: Experiment on MNIST-Split-5x2 using regularization terms $5 \cdot 10^{-3}$

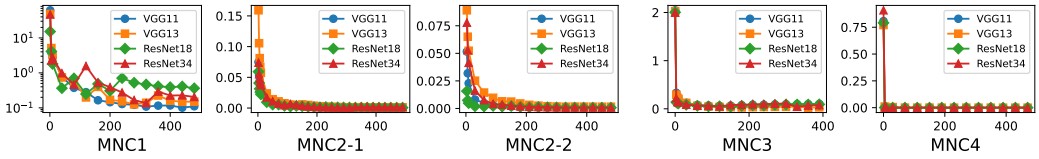

Figure 49: Experiment on MNIST-Split-5x2 using regularization terms $5 \cdot 10^{-5}$

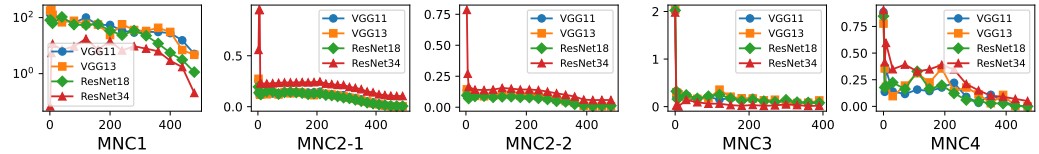

Figure 50: Experiment on CIFAR10-Split-5x2 using regularization terms $5 \cdot 10^{-3}$

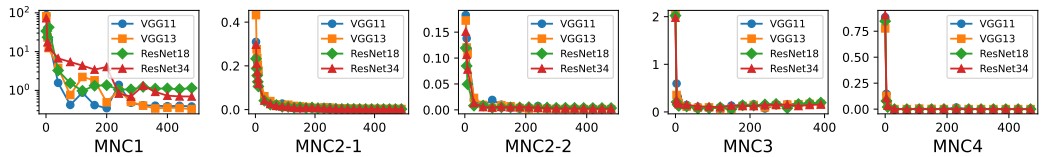

Figure 51: Experiment on CIFAR10-Split-5x2 using regularization terms $5 \cdot 10^{-5}$

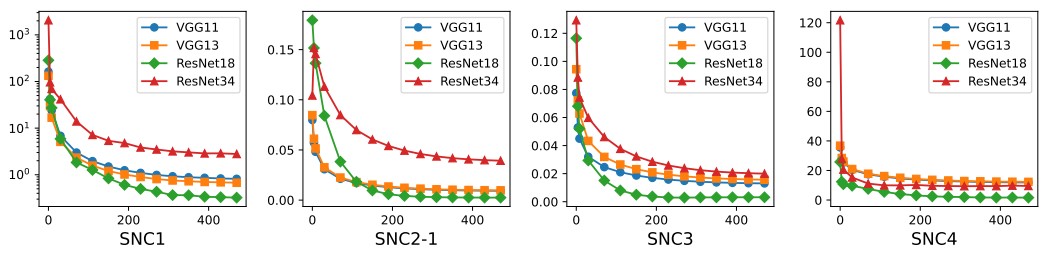

Figure 52: Experiment on Multi-MNIST using learning rate $1 \cdot 10^{-2}$

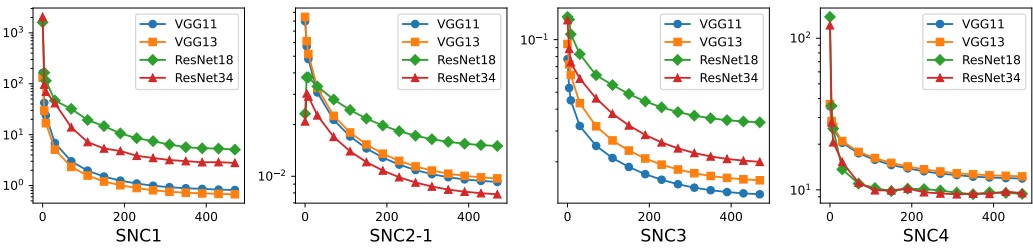

Figure 53: Experiment on Multi-MNIST using learning rate $1 \cdot 10^{-3}$

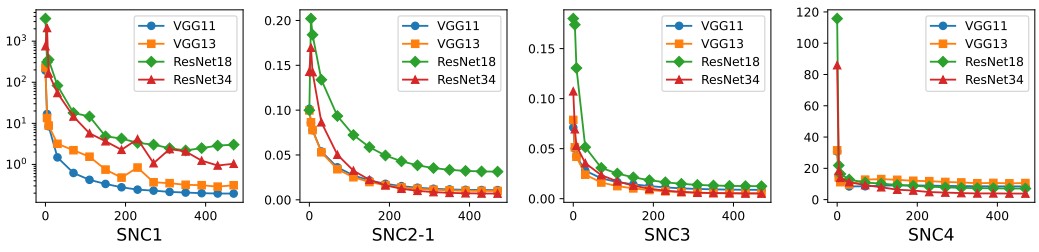

Figure 54: Experiment on Multi-CIFAR10 using learning rate $1 \cdot 10^{-2}$

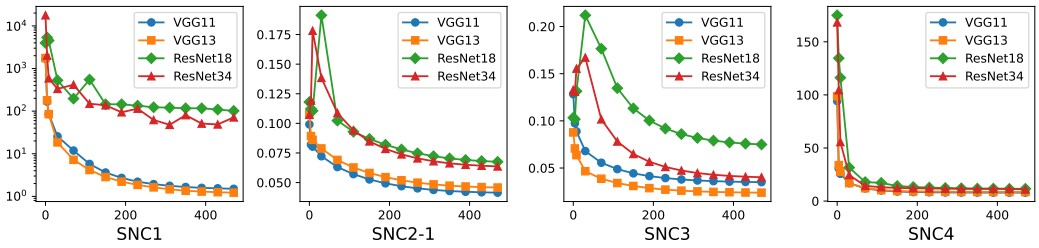

Figure 55: Experiment on Multi-CIFAR10 using learning rate $1 \cdot 10^{-3}$

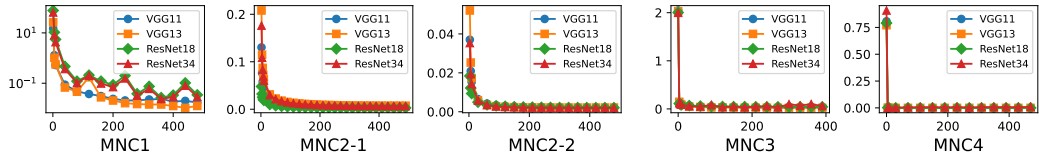

Figure 56: Experiment on MNIST-Split-5x2 using learning rate $1 \cdot 10^{-2}$

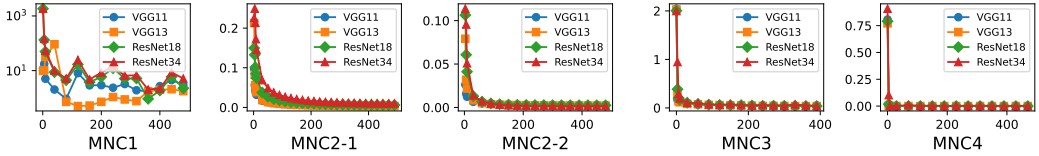

Figure 57: Experiment on MNIST-Split-5x2 using learning rate $1 \cdot 10^{-3}$

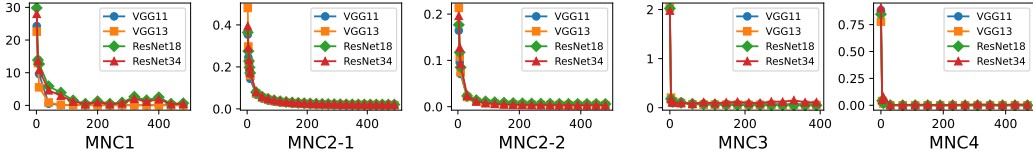

Figure 58: Experiment on CIFAR10-Split-5x2 using learning rate $1 \cdot 10^{-2}$

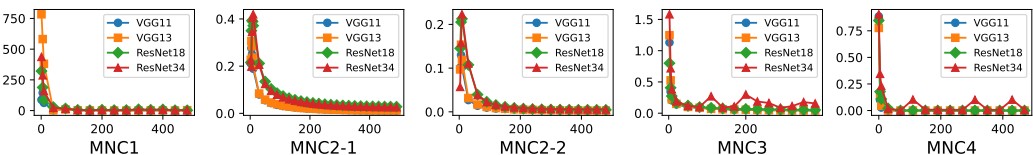

Figure 59: Experiment on CIFAR10-Split-5x2 using learning rate $1 \cdot 10^{-3}$

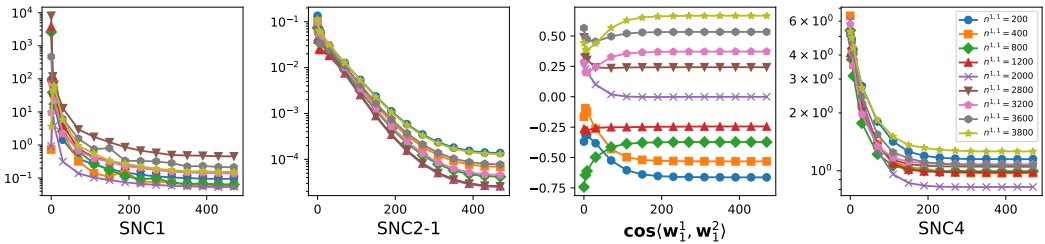

Figure 60: Experiment on Multi-MNIST ($K = 2$, $N = 8000$).

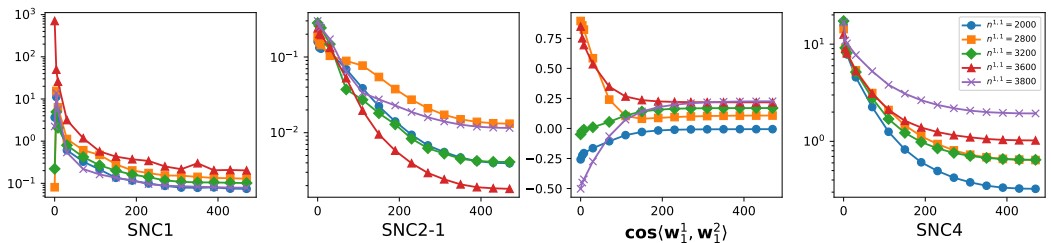

Figure 61: Experiment on Multi-MNIST ($K = 3$, $N = 18000$).

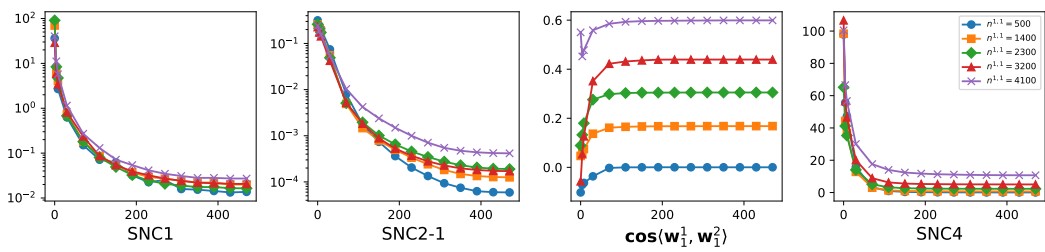

Figure 62: Experiment on Multi-MNIST ($K = 10$, $N = 50000$).

## D.7 EXPERIMENTS ON EFFICIENT TRAINING UTILIZING NEURAL COLLAPSE PROPERTIES

By leveraging the characteristics of neural collapse, modifications can be made to the multi-task network structure by reducing the dimension of the features to the number of classes, achieving parameter savings without compromising performance. Given the properties of SSMTC-NC and MSMTC-NC, we initialize the task-specific classifiers as orthogonal Simplex ETFs. We reduce the feature dimension to the sum of the number of classes for each task, and freeze the parameters of the last layer in the deep neural network during training.

As shown in Table 3, the experimental results demonstrate that we achieve parameter savings without compromising model performance by utilizing the ETF classifier. When applied to CIFAR100-Split-5x20, better performance can be achieved. For example, we achieve the $25.74\%$ parameter saving when using the VGG11 network for classifying the CIFAR100-Split-5x20 dataset.

Since the NC behavior in MSMTC closely resembles that in single task learning, it naturally suggests that methods developed for NC under imbalanced single-task learning could also be effective for imbalanced MSMTC. Following Yang et al. (2022), the final layer is replaced with a Simplex ETF and dot-regression (DR) loss is adopted to improve accuracy under class imbalance. We evaluate the approach on CIFAR-10 and CIFAR-100 with an imbalance ratio of 0.01. As shown in Table 4, this strategy leads to consistent performance gains in imbalanced MSMTC, indicating that insights derived from NC in single task learning could be transferable to the MSMTC setting.

## D.8 EXPERIMENTS ON GENERAL MTL

When no assumptions are made about task distribution and class, we find that features of the same class collapse together as illustrated in Figure 63.

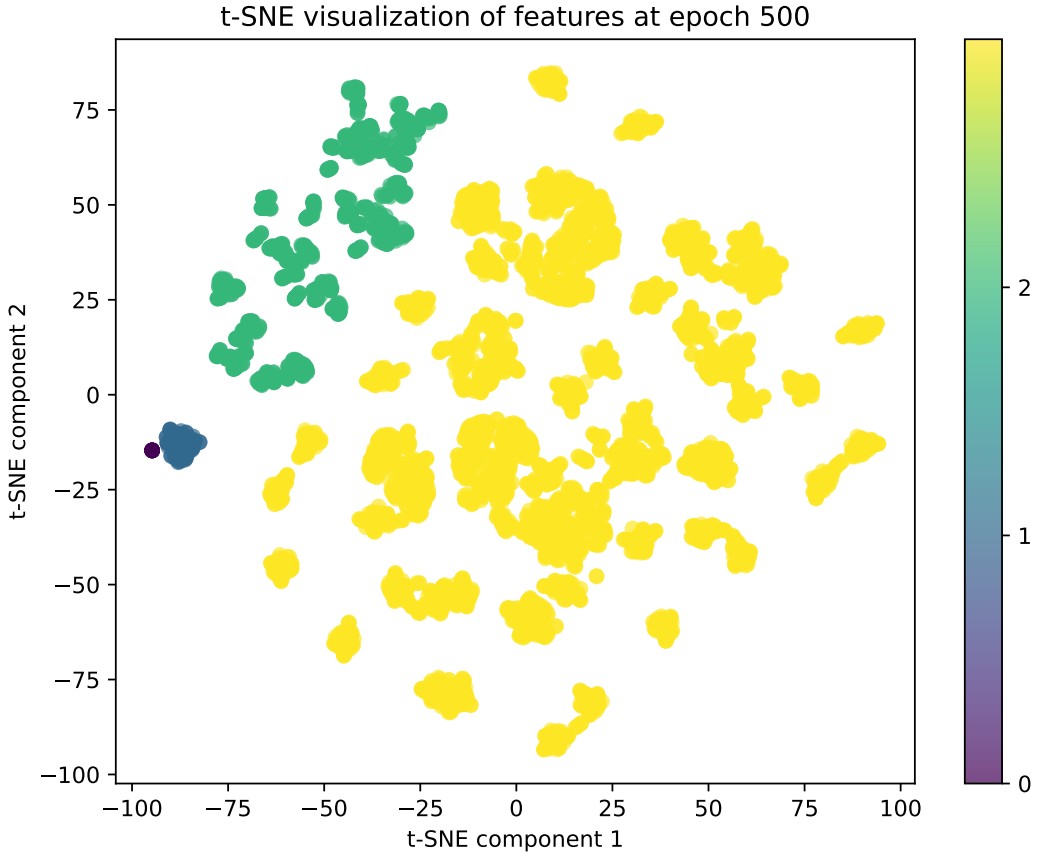

Figure 63: Experiment on Multi-MNIST

We change the sample size for each class from 500 to 10 and found that neural collapse still exists, as illustrated in Figure 64 and Figure 65.

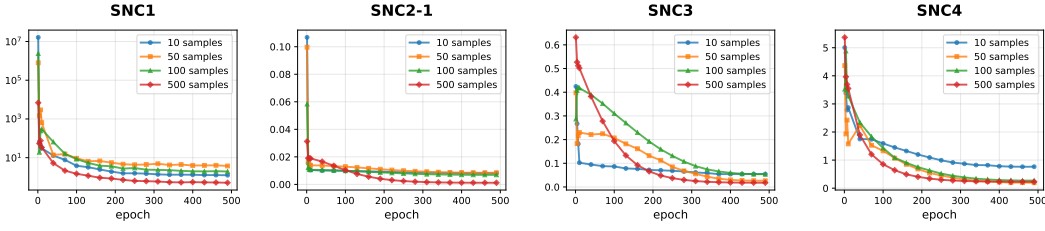

Figure 64: Experiment on Multi-MNIST

## D.9 EXPERIMENTAL METRICS

Table 5 and Table 6 show the metrics of verify SSMTC-NC and MSMTC-NC respectively.

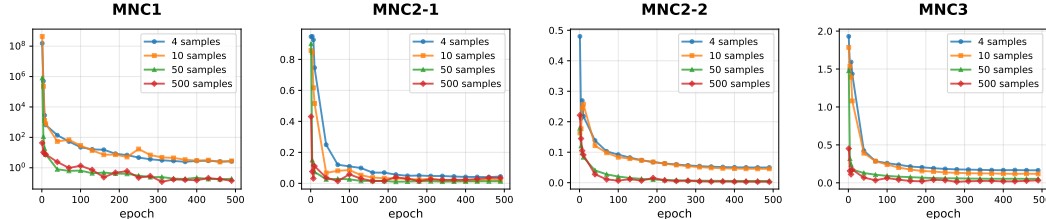

Figure 65: Experiment on MNIST-Split-5x2

Table 3: Performance and parameter saved using learned and ETF classifier

| Dataset Arch | ResNet18 | | ResNet34 | | VGG11 | | VGG13 | |
| --- | --- | --- | --- | --- | --- | --- | --- | --- |
| | Learned | ETF | Learned | ETF | Learned | ETF | Learned | ETF |
| **Test Accuracy (%)** | | | | | | | | |
| CIFAR100-Cross-10x10 | 53.32 | 53.52 | 56.58 | 57.16 | 55.68 | 55.54 | 59.94 | 60.27 |
| MNIST-Split-5x2 | 99.90 | 99.83 | 99.81 | 99.78 | 99.68 | 99.78 | 99.85 | 99.75 |
| CIFAR10-Split-5x2 | 90.44 | 91.22 | 92.04 | 91.82 | 90.61 | 91.43 | 93.38 | 93.46 |
| CIFAR100-Split-5x20 | 62.45 | 65.20 | 60.03 | 61.90 | 68.82 | 69.76 | 69.07 | 70.53 |
| **Parameter Saved (%)** | | | | | | | | |
| CIFAR100-Cross CIFAR100-Split | 0 | 21.26 | 0 | 11.18 | 0 | 25.74 | 0 | 25.24 |
| MNIST-Split CIFAR10-Split | 0 | 20.71 | 0 | 10.87 | 0 | 25.13 | 0 | 24.63 |

# E DISCUSSION ABOUT THE RELATED WORK

## E.1 RELATED WORK ABOUT NEURAL COLLAPSE

**The difference between Li et al. (2024) and ours**.

- **Analytical Focus**. Li et al. (2024) primarily investigates neural collapse patterns under multi-label settings and implications on guiding multi-label training. Our work mainly focuses on the geometry of task-specific classifiers and features learned by each task, and investigates how task correlation impacts features learned by each task.

- **Experiment Difference**. Our experimental setup differs substantially from that of Li et al. (2024), encompassing the network architecture, loss function, and dataset construction. The network in Li et al. (2024) includes a single classifier layer, while our work employs a task-specific classifier layer for each task.

- **Theoretical Contribution**. Whereas Li et al. (2024) establishes theoretical guarantees for NC in multi-label classification, our work provides the first formal characterization of NC phenomena in MTL frameworks, incorporating heterogeneous task weights, class-imbalanced scenarios, and task-correlation-driven geometric transformations of task-specific classifier manifolds. Previous research never required consideration of alignment between task-specific classifiers and features. In their UFM proofs, $h$ and $w$ maintain a one-to-one correspondence, expressed as $h = \lambda w$, where $\lambda$ is the same for each class. In the SSMTC setting, we need to consider the weighting coefficients of features learned by different task-specific classifiers within the shared feature representation as shown in (32). These coefficients are influenced by the number of classes per task, the task weights assigned to each task, and the task correlation. Only with properly configured coefficients can the equality conditions of a series of inequalities hold simultaneously, thereby ensuring that the optimal solution is attainable.

$$h = \lambda_1 w_1 + \lambda_2 w_2 + \cdots + \lambda_T w_T \tag{32}$$

Table 4: Long-tailed classification accuracy (%).

| Dataset Arch | ResNet18 | | ResNet34 | | VGG11 | | VGG13 | |
|---|---|---|---|---|---|---|---|---|
| | Learned | ETF+DR | Learned | ETF+DR | Learned | ETF+DR | Learned | ETF+DR |
| CIFAR10-Split-5x2 | 73.47 | 77.21 | 74.92 | 78.29 | 73.32 | 76.62 | 74.81 | 79.12 |
| CIFAR100-Split-5x20 | 53.24 | 56.61 | 54.13 | 57.21 | 53.76 | 55.87 | 54.48 | 56.13 |

Table 5: SSMTC-NC Metrics

SNC1 $\quad \underset{k_1...,k_T,j}{\mathbf{Ave}} \{ (\boldsymbol{h}_j^{k_1...,k_T} - \boldsymbol{\mu}^{k_1...,k_T})(\boldsymbol{h}_j^{k_1...,k_T} - \boldsymbol{\mu}^{k_1...,k_T})^\top \}$

SNC2-1 $\quad \underset{k,k' \neq k,t}{\mathbf{Ave}} \{ |\mathbf{cos}(\boldsymbol{w}_k^t, \boldsymbol{w}_{k'}^t) + \frac{1}{K-1}| \}$

SNC2-2 $\quad \underset{t}{\mathbf{Ave}} \{ \mathbf{Std}(\{\|\boldsymbol{w_1^t}\|_2, \|\boldsymbol{w_2^t},\|_2, \ldots, \|\boldsymbol{w_K^t}\|_2\}/\mathbf{Ave}\{\|\mathbf{w_1^t}\|_{\mathbf{2}}, \|\mathbf{w_2^t}\|_{\mathbf{2}}, \ldots, \|\mathbf{w_K}\|_{\mathbf{2}}\}) \}$

SNC3 $\quad \underset{k,k',t,t' \neq t}{\mathbf{Ave}} \{ \|\mathbf{cos}(\boldsymbol{w}_k^t, \boldsymbol{w}_{k'}^{t'})\| \}$

SNC4 $\quad \underset{k_1,k_2,\ldots,k_T,j}{\mathbf{Ave}} \{ \left\| \tilde{\boldsymbol{h}}_j^{k_1,k_2,\ldots,k_T} - \frac{\sum_{t=1}^T \boldsymbol{w}_{k_t}^t}{\|\sum_{t=1}^T \boldsymbol{w}_{k_t}^t\|_2} \right\|_2 \}$

SNC5 $\quad \frac{1}{N}\sum_{i=1}^N [(\underset{k_1}{\mathbf{argmax}}\langle \boldsymbol{w}_{k_1}^1, \boldsymbol{h}_i\rangle + b_{k_1}^1, \ldots, \underset{k_T}{\mathbf{argmax}}\langle \boldsymbol{w}_{k_T}^T, \boldsymbol{h}_i\rangle + b_{k_T}^T) \neq \underset{k_1...,k_T}{\mathbf{argmin}}\|\boldsymbol{h}_i - \boldsymbol{u}^{k_1...,k_T}\|_2]$

## E.2 RELATED WORK ABOUT MULTI-TASK LEARNING

In recent years, multi-task learning has witnessed substantial advancements (Liu et al., 2019b; Liu & Tsang, 2015; Mao et al., 2022; 2021; 2020a;b; Gong et al., 2023); however, research on its underlying mechanisms remains limited.

Shenouda et al. (2024) indicates that the norm associated with vector-valued variation spaces encourages the learning of shared features that are useful for multiple tasks. Our framework provides mechanistic insights through two key advancements. (1) We explicitly characterize shared features as a linear combination of task-specific classifiers. (2) Moreover, task-specific classifiers are aligned with the task-specific latent features, which are the features learned by each task. Our work provides the granular characterization of how shared features are useful for each task in modern deep neural networks.

Collins et al. (2024) indicates that multi-task training aligns points with the same label across tasks. Consistent with the theoretical framework, our experimental observations of NC1 further confirm that features sharing the same label collapse to their class mean.

Table 6: MSMTC-NC Metrics

| | |
|---|---|
| MNC1 | $\underset{t,k,j}{\mathbf{Ave}}\{(\boldsymbol{h}_{k,j}^t - \boldsymbol{\mu}_k^t)(\boldsymbol{h}_{k,j}^t - \boldsymbol{\mu}_k^t)^\mathsf{T}\}$ |
| MNC2-1 | $\underset{k,k'\neq k,t}{\mathbf{Ave}}\{|\mathbf{cos}(\boldsymbol{w}_k^t, \boldsymbol{w}_{k'}^t) + \frac{1}{K-1}|\}$ |
| MNC2-2 | $\underset{t}{\mathbf{Ave}}\{\mathbf{Std}(\{\|\boldsymbol{w_1^t}\|_2, \|\boldsymbol{w_2^t}\|, \|_2, \ldots, \|\boldsymbol{w_K^t}\|_2\}/\mathbf{Ave}\{\|\mathbf{w_1^t}\|_\mathbf{2}, \|\mathbf{w_2^t}\|_\mathbf{2}, \ldots, \|\mathbf{w_K^t}\|_\mathbf{2}\})\}$ |
| MNC3 | $\underset{k,t}{\mathbf{Ave}}\{\|\tilde{\boldsymbol{\mu}}_k^t - \tilde{\boldsymbol{w}}_k^t\|_2^2\}$ |
| MNC4 | $\underset{k,t,i}{\mathbf{Ave}}\{[\underset{k}{\mathbf{argmax}}\langle\boldsymbol{w}_k^t, \boldsymbol{h}_{k,i}^t\rangle + b_k^t \neq \underset{k}{\mathbf{argmin}}\|\boldsymbol{h}_{k,i}^t - \boldsymbol{u}_k^t\|_2]\}$ |

