# OpenReview forum: "Neural Collapse in Multi-Task Learning"
_ICLR.cc/2026/Conference — ICLR 2026 Poster_

### Official Review · Reviewer_ge6o · 2025-10-29

**Soundness:** 3
**Presentation:** 2
**Contribution:** 2
**Rating:** 6
**Confidence:** 3

**Summary:**

The manuscript discusses the phenomenon of Neural Collapse in the setting of multi-task learning. Specifically, it states that in two specific forms of multi-task learning with shared backbone and task-specific classification heads, adopting the unconstrained features model, neural collapse emerges. Besides theoretical results, also experiments on real datasets are provided.

**Strengths:**

+ theoretical insights into multi-task learning are a valuable addition to the neural collapse literature
+ the extensive experiments do a good job of confirming that NC does emerge in practice

**Weaknesses:**

While generally of high quality, I find the work to have some shortcomings in the presentation of the work and the strength of the scientific contribution.

* Presentation

For a predominantly theoretical paper, I found the presentation in several parts insufficiently precise or clear. For example, the main concepts SSMTC-NC (Section 3.1) and MSMTC-NC (Section 3.2) are introduces as the phenomenon that some quantities converge to some constants (often 0) in final phase of network training. However, the associated theorem state that any *global optimizer* of the training objectives *satisfies* SSMTC-NC (Theorem 3.1) and MSMTC-NC (Theorem 3.2), so there is no aspect of optimization or convergence.
The theorems are phrased without the constraints on the regularization strength that are usually required to prevent trivial solutions. The proofs (lines 764 and 943, respectively) correctly contain the statements that too strong regularization will lead to trivial instead of collapsed solutions.

The description of (theoretical) results would have benefited from giving more insights how the results are relevant and related, e.g. by discussing at least some border cases.
1) For $T=1$, all results should reduce to the known ones for (balanced, ...) NC with cross-entropy loss. This seems to be the case, but it would be good to explicitly state it.
2) For SSMTC with $T>1$, nothing seems to prevent that multiple tasks are *identical*. Take, e.g., $y_i^1=y_i^2=\cdots=y_i^T$ for all $i\in[N]$. In that case, the multi-task problem would reduce to identical copies of a single-task problem. $W^1=W^2=\cdots=W^T$ should be a globally optimal solution, and $H$ only needs to be of rank $K$ (which minimizes regularization cost). However this seems incompatible with NC3? The answer might be probably related to Theorem 5.1, where NC3 becomes Correlated-NC3, but the relation of assumptions is not clear to me. Potentially, more answers could be found in the proofs in the appendix, but given their repetitive nature, the manuscript's heavy notation and the too short time that ICLR provides for reviewing this year, I was not able to fully parse the proofs.
3) For MSMTC with $T>1$, each tasks has its own data and only the backbone is shared. But in the UFM, each task has its own  unconstrained features, which remove the central element of "hard parameter sharing" MTL. With independent features, the problem seems to become completely decoupled into independent classification tasks. Since we know that individual UFMs exhibit NC, what is there left to prove? And why would the result still be interesting?


* Scientific contribution

The submissions makes some conceptual, theoretical and experimental contributions. I do see that the conceptual contribution of establishing NC for MTL exist, but I do not find it particular strong or surprising, given the many settings in which final-layer NC has already been proven. The manuscript would benefit from a stronger discussion what benefit the community has from knowing that NC holds in the MTL-UFM setup. The theoretical contribution lies mainly in generalized definitions of NC, and the proofs of the main theorems. The latter mostly follow standard steps from single-task cross-entropy NC. There are new aspects, but e.g. I didn't notice any step in the proofs that I would expect to have broader impact outside of the scope of the work. The experiments are extensive and appear well done. I would encourage the authors to release their source code to allow for reproducibility. The results confirm collapse behavior for real data in many settings, but (as in previous studies) the difference between numeric training of real networks and properties of the global-optimizer in a UFM is quite big. Presumably, the solutions found numerically are not actual global minimizers, but how do they related? Some interpolating experiments could help to reduce this gap (e.g. numeric optimization of the UFM).


* Minor comments:
- reference (Yu et al, NeurIPS, 2020) is duplicate
- notation such as "5e-3" should better be $5\cdot 10^{-3}$
- line 724: "if we minus a vector/scalar" should be "if we subtract a vector/scalar"
- I found the line of argumentation in the proofs not always ideal to follow. You provide a number of inequalities, and extract conditions saying e.g. "The conditions under which equality holds are as follows:" (line 767). Usually, this would be a way to establish *sufficient* conditions, but for your results of the type "all global minimizers have a certain property", you need also *necessary* conditions. These follow e.g. from the "only if" part of Lemma A.2 (line 701), but such subtleties can easily get lost.

**Questions:**

I have some question that I hope will help me to understand the work better, in particular its contribution. This list includes the relevant question from the "weakness" section above:

1) For $T=1$, am I right that you recover the same NC properties as in single-task learning?

2) For Theorem 3.1, is it possible that tasks are identical? If yes, why wouldn't the features and weight matrices become identical, violating NC3?

3) Am I right that the UFM variant of MSMTC decouples the optimization tasks, so the "parameter sharing" aspect of MTL disappears? If yes, what insights into MTL do we still obtain?

4) is there any other (hidden) conditions beyond the bounds on the regularization strength requires for Theorems 3.1 and Theorem 3.2 to hold?

5) how is the setting of Theorem 5.1 not compatible with the assumptions of Theorem 3.1?

6) given the many prior results on NC, how will the community benefit from your result that NC emerges (also) in MTL

7) would you say that your proofs contain any steps of independent interest, i.e. besides their role in proving your theorems?

---

> ### Author Response · Authors · 2025-11-23
>
> Thank you very much for your detailed suggestions and responses to improve our work.
>
> **Q1**: When $T=1$, NC properties in SSMTC-NC and MSMTC-NC is the same and reduce to the known ones. You are right.
>
> **Q2**: Sorry for the misunderstanding caused. At the beginning of Section 2.2, we introduce the label in SSMTC. In Section 4.2, we assume that the label is balanced. For example, in the Multi-MNIST dataset ($K=2$), there are n samples labeled $(0,0)$, $(0,1)$, $(1,0)$, $(1,1)$, respectively, instead of n samples labeled $(0,0)$, $(1,1)$, respectively. The two tasks share the same input data but have different labels, making them fundamentally different.
>
> **Q3**: The UFM variant of MSMTC decouples the optimization tasks into multiple optimization tasks, but the "parameter sharing" aspect of MTL doesn't disappear.
> In our work on MSMTC, we mainly investigate whether the use of common feature extractors to extract features in multi task learning will affect NC, but we find that the impact on NC is not obvious. It indicates that the over-parameterized network can still achieve the ideal optimal solution in the MTL scenario.
>
> **Q4**: The condition for $\lambda_H$ and $\lambda_W$ in Theorem 3.1 is $\lambda _H\lambda_W<\frac{N}{4K}$. Theorem 3.2 the condition of $\lambda_H$ and $\lambda_w$ is $\lambda_H\lambda_W<\frac{n}{4}$. For the practicability of conditions, please refer to our reply to Q1 proposed by Reviewer vAKg.
>
> **Q5**: In Theorem 3.1, we assume that the numbers of samples labeled as $(k_1, k_2, \dots, k_T) (k_i \in [K])$ are all the same. Taking Multi MNIST ($K=2$) as an example, the numbers of samples labeled as $(0,0)$, $(0,1)$, $(1,0)$, and $(1,1)$ are the same. In Theorem 5.1, in order to investigate the impact of task correlation on NC, we set the condition of $K=2$, but relax the label-balanced condition. We set the labels within each task to be balanced, as shown in Figure 3(b).
>
> **Q6**: Our work mainly focuses on MTL compared to previous work on NC, mainly studying the features and classifier performance in multi task learning. Through further detailed analysis, we find that shared features are formed by the features learned by each task in SSMTC. Moreover, we reveal a fundamental inductive bias in MTL that task correlation reshapes the space of task-specific classifiers, thereby promoting alignment among the features learned by each task.
>
> **Q7**: Our theories proof presented in Theorem 3.1, Appendix B.2, and Theorem 5.1 all address the feature weighting mechanism in the SSMTC setting. Specifically, we consider the coefficients assigned to features learned by different task-specific classifiers within the shared representation, as expressed by $h^{k_1,k_2,\dots,k_T}=\lambda_1 w^1_{k_1}+\lambda_2 w^2_{k_2}+\dots+\lambda_T w^{T}_{k_T}$. These coefficients are influenced by several factors: the number of classes in each task, the task weights, and the inter-task correlations. Only when these coefficients are appropriately configured can the equality conditions of the relevant series of inequalities be simultaneously satisfied, thereby ensuring that the optimal solution is attainable. We think that the feature weighting mechanism is independent of contributions from other works.
>
>
> Thank you for your suggestions in the presentation. We have updated it in the newly uploaded revision. We welcome your valuable feedback and are open to any suggestions for further improvement.

---

> > ### Comment · Reviewer_ge6o · 2025-11-23
> > **Thank you, request for further clarifications**
> >
> > * Thank you for the clarifying response to Q2 and the related Q5. Just to confirm, when in the submission you write *training data is labeled for each label* you always mean that each vector-valued combination $(y^1_i,\dots,y^T_i)$ has the same number of training examples. I had not read it this way before, so probably others did not either. It would be good if you made this statement more explicit in Section 3.1, like you do for (5) and (6).
> > However, I find this assumption quite a lot stronger than the “balanced labels” assumptions in the single task NC works you cite for justification. The analog of those would be that for each task separately the labels are balanced. Asking for all combinations instead means that the number of training examples has to grow exponentially with the number of tasks. Also, if one has a new (even label-balanced) task to solve, one might not be able to add it to the MTL, as the labels of its training set would have to be compatible with the training sets of all other tasks.
> >
> > &nbsp;
> > * Regarding Q3, could you please clarify your reply? You write *The UFM variant of MSMTC decouples the optimization tasks into multiple optimization tasks, but the "parameter sharing" aspect of MTL doesn't disappear.*
> > I do not quite understand your claim here. In what sense is there still “parameter sharing” in the UFM result of Definition 2/Theorem 3.2? Are the minimizers of (4), and thereby their collapse properties, in any way different if I minimize the MSMTC-UFM (4) or if I solve the UFM for each task separately?
> > It appears that the rest of your answer discusses if a common feature extractor can achieve UFM-like behavior. That's an interesting question to study, but not related to NC in the UFM model, or Theorem 3.2 specifically.
> >
> > &nbsp;
> > * Regarding Q4, please add the constraints on $\lambda_H$ and $\lambda_W$ to the theorems, as otherwise these are technically incorrect.
> >
> > &nbsp;
> > * Regarding the presentation, I do not see much improvement. Citations are now formatted properly, and some comments were added, but the insufficient precision of writing has not changed. E.g. NC is still introduced as a convergence phenomenon, but then claimed as a property of the global optimum solution. I understand that the time is short and I do not expect a rewrite for the author response, but then do not claim that you updated your manuscript according to my suggestions.

---

> ### Author Response · Authors · 2025-11-24
>
> Thank you for your valuable feadback and patient guidance.
>
> **Q8**: We have update the statement about the condition like (5) and (6). We  agree that the assumption is somewhat strong, so in Section 5.2 and Appendix D.6, we consider weaker assumptions and only require that the labels within each task are balanced. Investigating Neural Collapse in more complex multi-task scenarios remains an avenue for our future research.
>
> **Q9**: Sorry for misunderstanding your question. In MSMTC, Theorem 3.1 and Definition 2 do not involve parameter sharing, only parameter sharing was used in the experiments.
>
> Theorem 3.1 states that we can replace the parameters of the last layer with Simplex ETF. In MSMTC, when each task is replaced with ETF, higher accuracy can be achieved when replaced with mutually orthogonal simplex ETFs, as shown in Table 3 in Appendix D.7, although we did not find any cases of classifiers being mutually orthogonal in MSMTC in our experiments.  This may indicate that orthogonality can reduce interference between tasks.
>
> **Q10**: Sorry for overlooking the issues about presentations. We have updated it in the newly updated revision.
>
> **Q11**: Sorry for overlooking the issues about presentations. We have updated the theorem in the manuscript in the newly updated revision.
>
> -----------------------------------------------------------------
> Sorry for the inconvenience caused by my reply. Thank you for your valuable feedback, which is very important to us.

---

> > ### Comment · Reviewer_ge6o · 2025-11-24
> > **request for final clarification**
> >
> > Dear authors,
> >
> > thank you for your replies. I now have a better impression of your work.
> >
> > Regarding Theorem 3.2 (which I take as the one you meant in Q9), I would like to return to my question from the original review: if the learning problem is completely decoupled with no sharing between tasks, how is your theoretical result still relevant for MTL? And, given that we know that each individual (single-task) UFM exhibits NC, what is there left to prove? In your response you again argue with later experimental observations, which might be inspired by the theory, but are in a different setting, which does not provide a justification for the formulation.
> >
> > Regarding the edits to the manuscript, unfortunately, I do not consider the new version much better. It now states properties as part of the theorems that are almost but not exactly the ones previously introduces as NC. What I meant in my comment is that the presentation of the work should be reconsidered to make it formally correct, concise and convincing, not just cosmetic edits. However, as mentioned before, I do not expect such a revision from an author response.

---

> > > ### Author Response · Authors · 2025-11-30
> > >
> > > Thank you for your response, and thank you as well for raising this important question. We would like to further clarify our results in the MSMTC setting.
> > >
> > > First, given that UFM treats features as freely optimizable variables [1][2][3][4], justified by the universal approximation theorem [5], the framework implicitly assumes that the model has sufficiently high representation capacity. In our analysis, we adopt the same assumption for multi-task models. Under this assumption, when the data for different tasks come from distinct sources (the MSMTC setting), the optimization problem decouples. Consequently, the shared model behaves as if there were no parameter sharing between tasks. We emphasize that this decomposition relies on two conditions simultaneously: (1) the multi-task model has high representation capacity, and (2) data are drawn from multiple different sources. Therefore, while this decomposition holds in MSMTC, it does not apply to SSMTC.
> > >
> > > As you have noted, after decoupling the optimization problem, we follow the same analytical approach as in STL and obtain results for MSMTC that closely resemble those in STL. We believe that these findings are both reasonable and meaningful. Our results clearly demonstrate that the NC phenomenon in MSMTC differs from that in SSMTC but closely aligns with NC in STL. Empirically, we observe across several real-world datasets that MSMTC indeed exhibits NC phenomena similar to those in STL, thereby validating our theoretical analysis. Furthermore, these results establish a clear connection between NC in STL and NC in MSMTC. This connection is meaningful as it suggests that insights developed from the NC phenomenon in the STL setting can be naturally extended to MSMTC, which is also supported by our experiments. For example,
> > >
> > >  (1) in the single-task setting, replacing the final layer with the simplex ETF improves parameter efficiency; similarly, in MSMTC, replacing the final layers with different Simplex ETFs is also observed to be effective, as shown in Table 3.
> > >
> > > (2) In the single-task setting, [6] demonstrates that employing an ETF classifier together with the dot-regression loss improves long-tailed classification performance. We extend this implication to the MSMTC setting and find that it continues to hold, as shown in Table 4 of the revised manuscript.
> > >
> > > We also would like to restate our contributions. The primary goal of this work is to investigate the NC phenomenon in multi-task learning. We study two widely used settings: SSMTC and MSMTC. In the SSMTC setting, Theorems 3.1 and 5.1 establish the existence of NC and demonstrate that its behavior differs significantly from NC in STL. In the MSMTC setting, Theorem 3.2 shows that NC also emerges, but its behavior closely resembles the STL case. The MSMTC setting is considered in our analysis to ensure that the investigation of NC in multi-task learning is comprehensive across both SSMTC and MSMTC.
> > >
> > > Therefore, while NC appears in both multi-task settings, its characteristics differ fundamentally between SSMTC and MSMTC. These findings further indicate that practical implications derived from the NC phenomenon should be considered separately for each setting. Finally, our experimental results consistently support and validate these theoretical insights.
> > >
> > >
> > > We are revising the presentation to achieve clearer and better expression. Thank you for your valuable suggestion and feedback. We would greatly appreciate any questions or suggestions you may have.
> > >
> > >
> > > [1] Neural collapse with unconstrained features.
> > >
> > > [2] Exploring deep neural networks via layer-peeled model: Minority collapse in imbalanced training.
> > >
> > > [3] Neural Collapse to Multiple Centers For Imbalanced Data.
> > >
> > > [4] Generalized Neural Collapse for a Large Number of Classes.
> > >
> > > [5] Multilayer feedforward networks are universal approximators.
> > >
> > > [6] Inducing neural collapse in imbalanced learning: Do we really need a learnable classifier at the end of deep neural network?

---

### Official Review · Reviewer_Y9is · 2025-11-01

**Soundness:** 4
**Presentation:** 3
**Contribution:** 4
**Rating:** 8
**Confidence:** 2

**Summary:**

This paper analyses Neural Collapse (NC) properties in Multi-Task Learning networks with single and
multiple sources (SSMTC and MSMTC). In the MSMTC scenario, the learned representations and task-specific classifier weight vectors converge to a unified simplex ETF geometry similar to STL. In the SSMTC scenario, each task's weights converges to a separate simplex ETF, and these are mutually orthogonal across tasks. They reveal some interesting properties concerning the
link between the task-specific classifiers as well as the relationship of the shared features and the task-
specific classifiers. Furthermore, they provide theoretical guarantees and extensive empirical validation across different network architectures, data sets and training conditions. Overall, this work extends NC analysis beyond the single-task case and offers valuable new, theoretically grounded insights for MTL.

**Strengths:**

The manuscript provides a non-trivial theoretical extension of Neural Collapse to multi-task learning. It is proven that any global optimum in both the SSMTC and MSMTC scenarios must satisfy analogous NC properties.

Extensive empirical evaluation is conducted, using multiple architectures, datasets, and multi-task loss weighting strategies. All of these choices are well-grounded in recent literature.

The findings reveal novel insights into geometric structures in how MTL networks organize their features. These insights deepen our understanding of how tasks interact in a shared model. This is important in a literature dominated by heuristic optimization (weighting) strategies without sound theoretical backdrop.

The paper shows that the degree of task relatedness can shape the learned feature geometry, which helps in treating tasks as heterogeneous and understanding when and why MTL works.

**Weaknesses:**

The discussion of practical implications is fairly limited. A better connection to practical MTL challenges could be made. For instance, it is unclear how these collaps properties might help improve training procedures. Linking the insights from this paper to issues such as task imbalance, negative task transfer and MTL architecture design would be very valuable.

**Questions:**

Do the observed NC structures also manifest on hold-out data? Can you say something about how NC relates to generalization in this context? In STL, NC has been linked to improved generalization. Any insight into how NC on training data translates to performance or structure on test data would be appreciated.

In SSMTC, the orthogonality between task-specific classifier subspaces seems elegant. In reality, deep features are typically shared across tasks. Do you view this orthogonality as a theoretical idealization, or do you think partial orthogonality might actually be beneficial in practice (e.g., for reducing task interference)?

Do any of the differences observed between MTL and STL settings suggest concrete benefits of training tasks simultaneously (as done here) versus training the same tasks independently on separate models?

The theoretical proofs rely on the Unconstrained Feature Model (UFM) and the existence of a global optimum. In practice, deep networks rarely reach global optima. How sensitive are your predicted NC structures to this? For instance, does partial or approximate collapse occur under suboptimal convergence?

Your linear combination view seems to suggest that the shared representation in MTL is fundamentally additive. Do you agree with this?

**Details Of Ethics Concerns:**

No concerns

---

> ### Author Response · Authors · 2025-11-23
>
> Thank you very much for your detailed suggestions and responses to improve our work.
>
> **Q1** and **Q2**: In the test data, NC in MTL still exists, but the NC1 properties in the test data are not as high as the degree of collapse of NC1 in the training data. We found that in MSMTC, when each task is replaced with ETF, higher accuracy can be achieved when replaced with mutually orthogonal simplex ETFs, as shown in Table 3 in Appendix D.7, although we did not find any cases of classifiers being mutually orthogonal in MSMTC in our experiments. So, we think that when two tasks are irrelevant, the orthogonality between task-specific classifier subspaces might actually be beneficial for reducing task interference in practice.
>
> **Q3**:
> 1. MTL uses fewer parameters compared to training T individual models
>
> 2. A single STL model can also achieve NC and form a Simplex ETF. However, the orientation of this framework in the feature space is arbitrary. The feature coordinate systems learned from different tasks are completely misaligned and lack a unified semantic interpretation. In SSMTC-NC, the shared features are composed of single task features, which provide a unified and intrinsically aligned semantic coordinate system.
>
> **Q4**: We conducted a large number of experiments using various datasets, multiple multi task optimization methods, and multiple regularization parameters, indicating that NC in MTL is a relatively robust existence as shown in Appendix D. When the global optimum cannot be achieved due to optimization issues, features of the same class will still collapse together, but the relationship between features and classifiers will become uncertain.
>
> **Q5**: Yes. We employ the equation $h^{k_1, k_2, \dots, k_T}=\lambda (w^1_{k_1}+w^2_{k_2}+\dots, w^T_{k_T})$ to elucidate the additive nature of the shared feature representation. This formulation provides a theoretically grounded explanation for the alignment between features of individual tasks and the final shared features. Specifically, when a task is well-learned, feeding the network a sample from only the $k$-th class of the $t$-th task results in a feature representation that aligns with its corresponding classifier vector ${w}_{k}^t$. This principle directly explains our empirical observation on MultiMNIST: the feature vector of a complete image (containing multiple digits) equals the sum of the feature vectors obtained from its individual components (each containing a single digit).
>
> We welcome your valuable feedback and are open to any suggestions for further improvement.

---

### Official Review · Reviewer_vAKg · 2025-11-06

**Soundness:** 3
**Presentation:** 2
**Contribution:** 3
**Rating:** 6
**Confidence:** 3

**Summary:**

This work formalizes two settings, SSMTC and MSMTC, under the unconstrained feature model and prove that any global minimizer exhibits NC geometry extended to multi-task, introducing new NC measurement, SNC1–SNC5 for SSMTC and MNC1–MNC4 for MSMTC, to quantify convergence to the predicted NC geometry. Experiment is verified over multiple backbone and datasets. To my knowledge, the formulations should be new, and the contribution over decorrelation across tasks & correlation-aware alignment is also supposed to be new.

**Strengths:**

- The overall work is well written. The setup and background sufficiently familiarize readers with the problem.
- I briefly reviewed the appendix proofs and believe the main claims and high-level theoretical results are sound. Some clarity issues remain; please see my questions.
- The experiments are extensive across different setups and backbones and are presented under multiple MOO weighting strategies. I believe most of the claims are verified.
- As noted in the summary, the theoretical contribution is new and the derivations appear solid (I did not spot any major errors). However, the technical novelty is somewhat limited, as the work reads as an extension of single-task neural collapse rather than a new paradigm.

---

Overall, I believe this is a new contribution in theory towards multi-task NC, even though it might be somehow incremental to view from technical theoretical contribution. My current assessment is boardline; I will re-evaluate it based on the rebuttal. Still, please address the concerns in weakness section.

**Weaknesses:**

- The method assumes balanced labels. I suggest to add discussion of how label imbalance and skewness would affect your modules and final results, through some specific quantitative analysis.

- Following from the previous point, although the balanced-label assumption aligns with prior work, the datasets used are relatively standard and small-to-medium for probing NC geometry. The effect on large, real-world datasets remains unclear and is therefore a limitation. Please consider adding at least one larger split to demonstrate robustness for your readers. For the medium-size experiment in Appendix D.4, report mean & std in the curve across multiple seeds. I also suggest moving the real-dataset experiments on ImageNet and CelebA from the appendix into the main paper, replacing some synthesized-data experiments.

- Several constraints are hard to interpret in terms of realistic, practical impact, and I have concerns about the UFM setup. See Questions for details.

- The clarity of writing and presentation should be improved. See Questions for details.

**Questions:**

- For thm3.1/3.2. All the NC results only sound given nontrivial $\lambda_{H}\lambda_{W}$ less than the threshold, but as you derived in line 764, when $\lambda_{H}\lambda_{W}$ is bigger than the threshold, the global minimizer is simply a trivial solution, making the NC properties no longer hold. How should I interpret this in terms of actual applicability? I suggest to add several real examples of $N$ and $K$ from your actual dataset to justify to what extent such condition would hold.

- For the whole UFM setup, I am concerned that the actual network you trained is not optimizing the UFM objective you analyzed, which should be a deep shared feature extractor plus per-task heads with CE. The UFM analysis in multi-task form breaks the coupling between tasks, which seems like a toy model.


Presentations:

- Please properly use \citet for better readability, e.g., line 181, line 182-183, and all the citations from Appendix E.

- Please consider to add all eq ref in your proof. Some of them are not added, and it reads somehow inconvenient, e.g., line 954 ref to inequality m.

- Please consider add a table to present all the metrics in section 4.3.

- typo. Line 114, SSTMC-NC

---

> ### Author Response · Authors · 2025-11-23
>
> Thank you very much for your detailed suggestions and responses to improve our work.
>
> **Q1**: About the condition $\lambda_H \lambda_W$
>
> **A1**: $\lambda_W$ are normally set to decimals less than 0.05 while $\lambda_H$ is smaller in practice. under the most extreme conditions, $K=2$, each class has $1$ sample, which also meets all the conditions in the theory. We vary the sample size for each class to verify the neural collapse and found that it still exists.  We have added this part of the experiment in Appendix D.8 of the newly uploaded revision.
>
> **Q2**: About the limitation of UFM.
>
> **A2**: Thank you for raising this important point about the UFM setup. We use UFM to analysis the training loss to get the optimal solution to reach the least loss. Although UFM does not analyze neural network optimization from the perspective of training momentum, let me explain from the loss perspective that the network structure we analyzed is indeed the network structure we trained.
> In MSMTC, during network training, although each loss term may be affected by conflict, optimization is aimed at achieving the minimum weighted total loss, and minimizing each single task loss can always achieve the minimum total loss. In SSMTC, we use $h^{k_1, k_2, \dots, k_T} _i$ to represent the features of the $i$-th sample with label $(k_1, k_2, \dots, k_T)$. The first task is to classify $h^{k_1, k_2, \dots, k_T} _i$ as $k_1$, and so on. When analyzing SSMTC loss, it is not actually divided by tasks, but by each sample in the theory, so in this setting, tasks are related experimentally and theoretically.
>
> Thank you for your suggestions in the presentation. We have updated it in the newly uploaded revision. As for more experiments, we are still improving and striving to supplement as soon as possible. We welcome your valuable feedback and are open to any suggestions for further improvement.

---

> > ### Comment · Reviewer_vAKg · 2025-11-25
> >
> > For Q1, I am getting a better sense of the hyperparameters. I can confirm this is resolved.
> >
> > For Q2, I might need more time to understand your explanation, but I feel at least in higher-level, it makes sense. Please consider to add this to the appendix if you haven't done so. Readers might need this to get a better understanding regarding the validity of UFM setup.
> >
> > Presentation issues, e.g., citations, eq ref, I raised in my original review is also fixed.
> >
> > No pressure on the experiment if you have no time to complete the one on large split. It only minds to me whether the paper is further strengthened, and I won't interpret it as the weakness to reject this paper even if you cannot complete it.
> >
> > Given the weaknesses resolved, I am more confident with my positive score and will maintain it, leaning towards accept.

---

> > > ### Author Response · Authors · 2025-11-30
> > >
> > > We are pleased to acknowledge that we have partially addressed the reviewer's concerns. Although we have provided an explanation of the UFM setup, we understand that some aspects may still remain unclear. Therefore, we further clarify how the UFM formulation applies to both SSMTC and MSMTC.
> > >
> > > First, in the UFM framework, features are treated as freely optimizable variables, which is justified by the universal approximation theorem [1]. This implicitly assumes that the model possesses sufficiently high representation capacity, regardless of whether parameters are shared across tasks. We understand the reviewer's concern regarding whether parameter sharing might effectively disappear under this assumption, thereby breaking the coupling between tasks.
> > >
> > > It is important to emphasize that in our analysis, the model is still a single multi-task model rather than multiple separate models. In the SSMTC setting, all tasks share data drawn from the same source, and thus the coupling between tasks remains explicitly preserved through shared features. In contrast, in MSMTC, the optimization objective can indeed be decomposed into a sum of single-task objectives. This decomposition reflects that the effect of parameter sharing becomes negligible, but it does not mean that the model becomes multiple independent models; the analysis is still conducted within a single multi-task model. Therefore, within the UFM framework, the influence of parameter sharing is inherently stronger in SSMTC than in MSMTC.
> > >
> > > We now clarify why the effect of parameter sharing disappears in MSMTC. The key reasons are twofold: (1) the multi-task model has high representation capacity (UFM model), and (2) the data are drawn from multiple distinct sources (the MSMTC setting). Under these two conditions, the shared features can freely adjust to each task independently, which allows the optimization problem to decouple into task-specific subproblems. This decomposition holds in MSMTC but not in SSMTC. As a result, the NC phenomenon observed in MSMTC closely resembles that in STL, which is theoretically reasonable and further supported by our empirical findings.
> > >
> > > Finally, although the NC phenomenon appears in both SSMTC and MSMTC, the different roles played by parameter sharing imply that the resulting NC behaviors differ significantly across the two settings. Our theoretical analysis, together with our empirical results, consistently demonstrates these distinctions.
> > >
> > > We hope that this clarification resolves the remaining concerns. We have add this clarification in the Appendix B. Thank you for your valuable suggestion.
> > >
> > > Thank you for the suggestion about the experiments. We have added larger-scale split experiments on ImageNet in the revised manuscript; results are provided in Appendix D.4.
> > >
> > >
> > > We would greatly appreciate any questions or suggestions you may have. Thank you for your valuable feedback and suggestions.
> > >
> > >
> > > [1] Multilayer feedforward networks are universal approximators.

---

### Official Review · Reviewer_MDbD · 2025-11-06

**Soundness:** 3
**Presentation:** 3
**Contribution:** 3
**Rating:** 6
**Confidence:** 4

**Summary:**

This paper investigates the phenomenon of neural collapse (NC) in multi-task learning (MTL), extending the existing understanding of NC beyond single-task settings. The authors propose two novel MTL-specific NC formulations—SSMTC-NC and MSMTC-NC—and provide both theoretical guarantees under the Unconstrained Feature Model (UFM) and extensive empirical validation across multiple datasets and architectures. They further explore how task correlation influences the geometry of task-specific classifiers and shared feature representations.

**Strengths:**

1.The paper is the first to systematically extend neural collapse to multi-task learning, introducing two well-defined geometric configurations (SSMTC-NC and MSMTC-NC).

2.The authors provide rigorous proofs showing that both SSMTC-NC and MSMTC-NC emerge as global optima under the UFM setting.

3.Experiments are conducted across a wide range of datasets (e.g., Multi-MNIST, CIFAR100-Split, CelebA) and architectures (ResNet, VGG), with additional studies on task weighting strategies and task correlation.

**Weaknesses:**

1. Most theoretical results assume balanced class and task distributions, which may not hold in practical MTL scenarios.

2. Key figures (e.g., Figure 1, 2) do not include confidence intervals or measures of variability, which could mislead readers about the robustness of results.

**Questions:**

1.How would the NC phenomena change in highly imbalanced MTL settings, both in terms of class and task distributions?

2.Have the authors considered evaluating the proposed NC metrics in few-shot or cross-domain MTL scenarios to assess generalizability?

---

> ### Author Response · Authors · 2025-11-23
>
> Thank you very much for your detailed suggestions and responses to improve our work.
>
> **Q1**: How would the NC phenomena change in highly imbalanced MTL settings, both in terms of class and task distributions?
>
>
> **A1**: When no assumptions are made about task distribution and class, we find that features of the same class collapse together. We add this in the newly uploaded revision in the Appendix D.8. As for the specific relationship between classifiers and features, it will be the focus of future research.
>
> When $K_1=K_2=K_3=... K_T$ is not satisfied, we have included this theory and experiments in the Appendix B.2 and Appendix D.2.
>
> In addition, in Section 5.2 and Appendix D.6, we conducted imbalanced experiments under certain constraints, hoping to provide some inspiration for future work
>
> **Q2**: Have the authors considered evaluating the proposed NC metrics in few-shot or cross-domain MTL scenarios to assess generalizability?
>
> **A2**: We vary the sample size to verify the neural collapse from 500 to 10 and found that it still exists. We have added this part of the experiment in Appendix D.8 of the newly uploaded revision.
>
> Thank you again for your valuable feedback to improve our work. We welcome your valuable feedback and are open to any suggestions for further improvement.

---

### Meta-Review · Area_Chair_jLvZ · 2026-01-06

**Summary:**

In this paper, the authors study the neural collapse (NC) phenomenon in multi-task learning (MTL), a setting that has not been systematically analyzed in prior work. The core contribution of the paper is a unified theoretical and empirical characterization of NC under two representative MTL settings: Single-Source Multi-Task Classification (SSMTC) and Multi-Source Multi-Task Classification (MSMTC). In particular, the authors show that in SSMTC, task-specific classifiers converge to task-specific Simplex Equiangular Tight Frames (ETFs), with the resulting task-wise ETFs being mutually orthogonal, while the shared features across tasks converge to the scaled sum of the corresponding task-specific classifier weights. In contrast, in MSMTC, the task-specific classifiers and features of each task converge to a Simplex ETF. Overall, this is a good paper.

(1) *Novelty and Technical Soundness*: This is the first work to systematically investigate neural collapse in the multi-task learning setting. The key novelty of the paper lies not merely in extending neural collapse to MTL, but in revealing that the collapse geometry is fundamentally shaped by task coupling, leading to distinct regimes that are absent in prior single-task analyses. This perspective provides a unified geometric understanding of how shared representations and task-specific classifiers interact in MTL. On the technical side, the theoretical analysis under the Unconstrained Feature Model rigorously characterizes the observed collapse phenomena as global minimizers of the training objective. During the discussion period, the authors further clarified modeling assumptions, explicitly stated parameter conditions in the theoretical results, and refined the presentation, which Reviewer vAKg noted as resolving their main concerns regarding regularization and hyperparameter conditions.

(2) *Empirical Evaluation and Practical Impact*: The empirical evaluation conducted across synthetic data, MultiMNIST, CIFAR-based task splits, and CelebA exhibits clear neural collapse behavior in both SSMTC and MSMTC settings. Reviewer Y9is particularly highlighted the breadth of datasets, architectures, and task-weighting strategies as a strength of the paper. In response to feedback from Reviewers MDbD and vAKg, the authors added additional experiments under relaxed balance assumptions, few-shot regimes, and larger-scale ImageNet splits, which significantly strengthened the empirical section.

Altogether, the paper makes a clear and novel contribution to understanding representation geometry in multi-task learning by extending neural collapse theory to previously unexplored settings. Therefore, I recommend accepting this paper.

**Reviewer Concerns:**

Please refer to the summary.

**Reviewer Scores:**

Please refer to the summary.

---

### Decision · Program_Chairs · 2026-01-26

Accept (Poster)